



# Multi-tool dataset on Northern Eurasian Riverbank Migration (NERM)

Sergey R. Chalov[1], Victor Ivanov[1], Danila Shkolnyi[1], Ekaterina Pavlyukevich[1,2], Michal Habel[3], Dmitry Botavin[1], Aleksandra Chalova[1], Pavel Golovlev[1], Arseny Kamyshev[1], Roman Kolesnikov[5], Uliana Koneva[1], Anna Kurakova[1], Nadezda Mikhailova[1], Elizaveta Tuzova[1], Kristina Prokopeva[1,2], Aleksandr Zavadsky[1], Rituparna Acharyya[3], Roman S. Chalov[1], Aleksandr Varenov[4], Leonid Turykin[1], Anna Tarbeeva[1], Daidu Fan[6]

[1] Lomonosov Moscow State University, Moscow, 119991, Russia
[2] Water Problems Institute of RAS, Moscow, 119333, Russia
[3] Uniwersytet Kazimierza Wielkiego, Bydgoszcz, 85-033, Poland
[4] Minin Nizhny Novgorod State Pedagogical University, Nizhny Novgorod, 603005, Russia
[5] Arctic Research Center of the Yamal-Nenets Autonomous District, Salekhard, 629008, Russia
[6] Tongji University, Shanghai, 200092, China

Correspondence to: Ekaterina Pavlyukevich (ekaterina.kornilova.hydro@gmail.com)

**Abstract.** Riverbank erosion monitoring and modeling has a long-standing tradition in Earth system science. The current body of research primarily relies on observations at the basin and reach levels. We endeavored to compile a comprehensive dataset of riverbank migration observations using a variety of measurement techniques, both field-based and remote sensing data. The dataset comprises information from twelve extensive river basins situated in Northern Eurasia, encompassing rivers that drain into the Baltic Sea, the Arctic and Pacific Oceans, and the Caspian Sea, specifically the catchments of the Vistula, Volga, Ural, Ob, Nadym, Yenisey, Lena, Indigirka, Yana, Kolyma, Amur and Kamchatka rivers. The rivers included in the dataset vary in terms of environmental conditions and have average discharges of between 0.3 and 19,700 $m^3$/s. This study examined approximately 140,000 kilometers of rivers in Northern Eurasia, covering small, medium, and large rivers, with data from up to 70 years of water classifications obtained from satellite images, including those from LandSat and Keyhole, across 626,700 river channel segments. The dataset collected average and maximum bank retreat rates (m/year), average areas of bank retreat ($m^2$/year), and volumes of channel erosion (t/year). It also recorded possible causes, encompassing both hydrological and catchment factors like permafrost, natural land zones, and geology. Our study showed that river discharge and permafrost distribution are the primary indicators of riverbank erosion in Northern Eurasia. These data will enhance the comprehension of bank erosion processes and their underlying factors, thereby facilitating the development of more accurate predictive models of river channels. The dataset is available open access via the ZENODO repository (https://doi.org/10.5281/zenodo.11072919) (Chalov et al., 2025).



# 1 Introduction

Approximately 2.5 billion individuals globally reside near major rivers and utilise them for water supply, transportation, and power generation (Musie and Gonfa, 2023) Characteristics of rivers change over time and in different locations, as noted in

(Knighton, 2014) and (Bracken et al., 2015). Specifically, alterations in their spatial limits are associated with hydromorphological processes occurring at various spatial scales, such as bank-scale, reach-scale, and floodplain-scale, and involve vertical and horizontal modifications of river channels (Alabyan and Chalov, 1998). The latter results in the most hazardous river-related phenomena, specifically lateral (riverbank) erosion, which leads to land loss and the conversion of floodplains into active channels.

Riverbank erosion has long been viewed as a threat to structures, engineering projects, infrastructure, and agricultural operations. This phenomenon can significantly contribute to sediment loads in rivers, thus serving as a crucial factor in sediment flux models (Kronvang et al., 2013; Wilkinson et al., 2009). Long-lasting effects on riverbank migration are caused by both natural and man-made changes to water and sediment supply, with the river's channel shape adjusting to new conditions (Alexeevsky et al., 2013; Brandt, 2000). Riverbanks serve as both a source (by erosion) and reservoir (by deposition on them)

for sediment, highlighting the effects of fluctuating sediment supply (Kronvang et al., 2013). Furthermore, chemicals such as metals and carbon stored in riverbank sediments are transported downstream to coastal seas due to bank deterioration (Reid and Dunne, 2016), in some cases significantly impacting terrestrial flux (Chalov and Ivanov, 2023; Chalov et al., 2023b; Gautier et al., 2021).

Researchers globally have investigated river planform transformations over the course of time to identify evolutionary patterns,

evaluate influencing factors, and control the fluvial ecosystem, which has led to a variety of measurement techniques. Laboratory flume experiments and detailed field measurements with erosion pins have validated methods for measuring river movement at reach scales  (Guy et al., 1966; Thorne, 1981). The conventional method of measuring bank erosion involves a thorough historical examination of riverbeds, taking into account available cartographic records and aerial images to evaluate changes in the river's shape (Fuller et al., 2003; Mandarino et al., 2019), which encompasses georeferencing of images,

interpreting photos, digitizing morphological features, and performing vector and raster geospatial analysis. Remote sensing data, including repeated LiDAR and optical remote sensing, notably improves the ability to track channel dynamics over large spatial areas and at decadal time intervals. These methods are based on retreat area detection, which involves assessing the movement of rivers by observing how channelized areas and regions without channels (like floodplains) evolve over time (Langhorst and Pavelsky, 2023). Comparisons of satellite images are used to create bank retreat polygons (Kurakova and

Chalov, 2019) or centerlines (Greenberg et al., 2023). Current methods of measuring river mobility through remote sensing focus mainly on meandering rivers with a single channel, and primarily emphasize bank movement (Donovan and Belmont, 2019; Sylvester et al., 2019). Furthermore, recent techniques like particle image velocimetry (PIV) (Chadwick et al., 2023) are being extensively utilized for monitoring bank erosion.



Studies on bank erosion are crucial, yet data on this topic are scarce due to the high expense of collecting and interpreting

them, and available data are mostly confined to specific river sections and watersheds. Datasets employed in riverbank erosion research include the GSWE – Global Surface Water Explorer, which uses a supervised classification of Landsat-5, -7, and -8 satellite imagery from 1984 to the present (Pekel et al., 2016), as well as the GLAD dataset, wherein water surface was calibrated with RapidEye imagery (Pickens et al., 2020). Global datasets designed to compile a consistent record of riverbank migration worldwide have been introduced by (Ielpi and Lapôtre, 2020), utilizing a sample of 983 meanders, and by (Langhorst

and Pavelsky, 2023), who created REAL (Riverbank Erosion and Accretion from Landsat) – a global dataset of riverbank erosion covering over 370,000 km of major rivers, based on GSWE and GLAD data.

These datasets are restricted to decadal-scale average riverbank erosion and rivers with widths exceeding 150 m, whilst also concentrating solely on the surface water occurrence dataset obtained from Landsat satellite imagery; as a result, more extensive observations of contemporary riverbank erosion rates are needed, encompassing different methods and rivers. In

Northern Eurasia, it is notably significant that nearly a quarter (about 26%) of areas with extremal bank erosion are recorded in the Langhorst and Pavelsky database (2023).

Recent studies on riverbank migration have been conducted for the biggest rivers in Northern Eurasia as part of international collaborative projects (Babiński et al., 2014; Lappalainen et al., 2018). Results of these studies, with rare exceptions (Alexeevskii et al., 2008; Alexeevsky et al., 2013), was not accessible to the international scientific community. The current

work enables the presentation of a multi-tool dataset of channel erosion rates encompasses rivers within the major catchments in Northern Eurasia, covering a total length of more than 140,000 km of river networks, also presented as an online GIS map (https://map.giscarta.com/viewer/93a6a4b3-179f-450f-be02-a31ca6db245b).

The purpose of this research is to provide a multi-scale dataset featuring outputs from the use of multiple tools to identify changes in planform at various scales, encompassing extreme values of bend migration at localized spots, estimates with a

spatial resolution of 10 to 100 m at the reach scale, and basin-scale averages with 1 km resolution. Following the dataset attributes description, we present a comparative analysis of contemporary riverbank erosion rates across different catchments and river sizes. We examine extreme variations in river channel changes across the entire dataset, focusing on large catchments such as the Ob River and specific environments like the Lena River delta, as well as long-term changes of bank erosion rates. The study of the dataset illustrates how it can be employed to calculate the in-channel contribution to sediment transport.

**2 Material and methods**

**2.1 Rivers**

This paper presents a dataset compiled from field-based observations and satellite images, including LandSat and Keyhole, that spans over 140,000 km of riverbanks in Northern Eurasia. The dataset covers small, medium, and large rivers and was compiled using different methods of image classification and digitization from bank transitions acquired over a 60-year

interval. The dataset comprises more than 626,000 river channel reaches, as detailed in Table 1, covering rivers belonging to

the watersheds of the Baltic Sea, Arctic and Pacific Ocean, and Caspian Sea. In total, over 250 rivers were examined across 28 distinct subdatasets, each of which focuses on a separate river section, or a group of rivers with similar landscape characteristics, or the largest rivers within a major watershed, or deltas as distinct distributary systems (Table 1). Each subdataset was derived through one of the methods detailed below and was identified by a unique ID within the dataset, ranging

from 1 to 28. The compiled data were employed in both statistical and qualitative examinations. The river distribution allows for the identification of the impacts of geology, hydrology, permafrost and vegetation by assigning each river to successive categories.

**Table 1. Description of rivers included in the dataset**

| ID | River | Catchment | L, km | $B_{mean}$, m/year | $B_{max}$, m/year | Q, $m^3$/s | Temporal range |
|----|-------|-----------|-------|---------|--------|---------|--------|
| 1 | Lena, Yakutsk | Lena | 1750 | 2.6 | 23.8 | 7272 - 7297 | 1965–1993 |
| 2 | Rivers of the Lena river basin | Lena | 34 800 | 0.8 | 14.6 | 30 - 15701 | 1986–2021 |
| 3 | Rivers of the Ob river basin | Ob | 30 200 | 0.5 | 7.67 | 30 - 13498 | 1985–2021 |
| 4 | Rivers of the Kolyma river basin | Kolyma | 8 100 | 0.81 | 8.84 | 30 - 3718 | 1999–2021 |
| 5 | Indigirka | Indigirka | 618 | 1.31 | 23.9 | 913-1,780 | 2000–2019 |
| 6 | Yana | Yana | 225 | 1.42 | 8.3 | 154-182 | 2002–2021 |
| 7 | Kolyma | Kolyma | 400 | 1.50 | 13.0 | 2,350-3,720 | 1965–2021 |
| 8 | Lena, delta | Lena | 14 600 | 0.89 | 5.99 | 15,800 | 2000–2021 |
| 9 | Rivers of the Yenisey river basin | Yenisey | 34 800 | 0.55 | 9.02 | 30 - 19744 | 1985–2019 |
| 10 | Ob | Ob | 4 000 | 1.67 | 26.3 | 1,040-13,400 | 1968–2022 |
| 11 | Ural | Ural | 1 700 | 2.18 | 20.9 | 64-337 | 1985–2015 |
| 12 | Sakmara | Ural | 330 | 1.90 | 8.3 | 44-250 | 1985–2015 |
| 13 | Setun and Ramenka | Volga | 20 | 0.15 | 0.62 | 1-2.4 | 1942–2010 |
| 14 | Rivers of Moscow region | Volga | 60 | 0.18 | 1.3 | 0.5-490 | 2003–2010 |
| 15 | Kamchatka | Kamchatka | 320 | 1.08 | 15.8 | 64-904 | 1967–2017 |
| 16 | Indigirka | Indigirka | 260 | 1.42 | 7.45 | 913-1,500 | 1975–2017 |
| 17 | Vistula | Vistula | 120 | 8.81 | 50.7 | 1,080-1,080 | 2006–2023 |
| 18 | Rivers of the Kudma river basin | Volga | 30 | 0.2 | 0.13 | 0.3-6.7 | 2017–2023 |
| 19 | Volga | Volga | 230 | 6.75 | 53 | 8,200-8,240 | 1977–2022 |
| 20 | Rivers of the Selenga river basin | Yenisey | 8 617 | 0.95 | 6.29 | 1-2,450 | 1984–2016 |
| 21 | Yenisey delta | Yenisey | 200 | 0.98 | 5.69 | 19,700 | 2000–2022 |
| 22 | Kolyma delta | Kolyma | 120 | 1.11 | 3.04 | 3,700 | 2001–2022 |
| 23 | Rivers of Yamal region | Pur, Nadym | 2 | 1.2 | 2.5 | 5-10 | 2022–2024 |



| 24 | Irtysh | Ob | 1 690 | 2.38 | 20.9 | 930-2,920 | 1985–2021 |
| 25 | Oka | Volga | 805 | 1.1 | 10.2 | 560-1,330 | 2002–2022 |
| 26 | Chulym | Ob | 1 100 | 1.69 | 13.2 | 240-780 | 2002–2022 |
| 27 | Ussuri | Amur | 214 | 1.06 | 8.5 | 210-820 | 2002–2022 |
| 28 | Messoyakha | Messoyakha | 296 | 2.17 | 14.5 | 100-310 | 1976–2023 |

Explanations: L – length of subdataset river sections [km]; $B_{mean}$ – average bank retreat rates [m/year]; $B_{max}$ – maximum bank retreat value [m/year]; $A_{mean}$ – average area of bank retreat [m²/year]; $Q_{mean}$ – annual mean river water runoff [m³/s] (Lehner, Grill, 2013).

The rivers presented in the dataset present contrasting conditions (Fig. 1) of channel erosion. The Vistula River (#17) case study reach extends over 120 km of the middle section of the Vistula River, between the mouth of the Radomka River and the Narew River (km 430-551 of waterway). This section is partially trained by transverse and longitudinal river groins (medium-water-level riverbed) and flood embankments (high-water-level riverbed). In the urban section, the riverbed is strongly confined, having been narrowed to embankments (e.g., in Warsaw), and incised (Bujakowski and Falkowski, 2019).

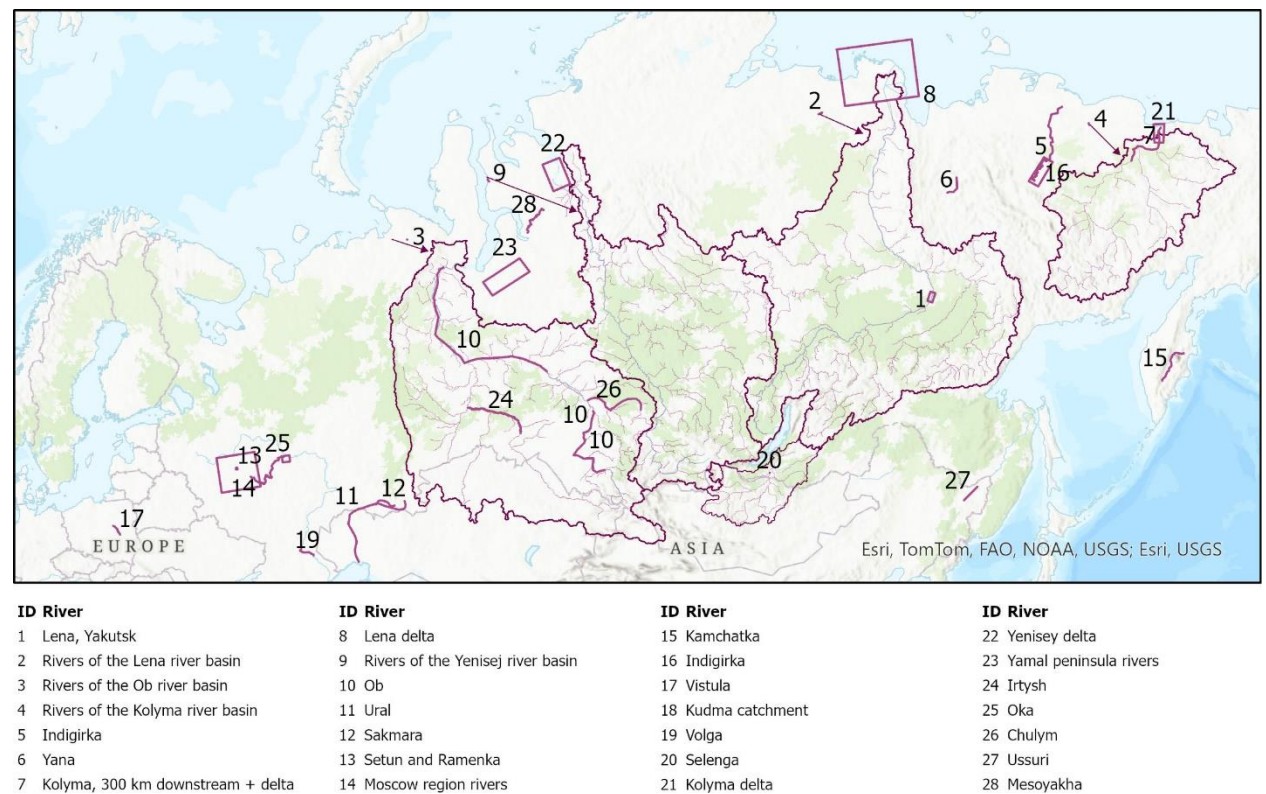

| ID | River | ID | River | ID | River | ID | River |
|---|---|---|---|---|---|---|---|
| 1 | Lena, Yakutsk | 8 | Lena delta | 15 | Kamchatka | 22 | Yenisey delta |
| 2 | Rivers of the Lena river basin | 9 | Rivers of the Yenisej river basin | 16 | Indigirka | 23 | Yamal peninsula rivers |
| 3 | Rivers of the Ob river basin | 10 | Ob | 17 | Vistula | 24 | Irtysh |
| 4 | Rivers of the Kolyma river basin | 11 | Ural | 18 | Kudma catchment | 25 | Oka |
| 5 | Indigirka | 12 | Sakmara | 19 | Volga | 26 | Chulym |
| 6 | Yana | 13 | Setun and Ramenka | 20 | Selenga | 27 | Ussuri |
| 7 | Kolyma, 300 km downstream + delta | 14 | Moscow region rivers | 21 | Kolyma delta | 28 | Mesoyakha |

**Figure 1. Locations of river sections included in the dataset**

The Volga River's largest right tributary, the Oka River (#25), is a vital waterway in the East European Plain, where a dense population has led to substantial human-induced pressure and notable changes in water and channel conditions. Sand mining



in riverbed quarries is a highly influential activity that can significantly impact the stability of a channel (Berkovich, Zlotina and Turykin 2023). The Oka River has a total length of 1500 km, with 805 km of that distance located in the middle and lower parts between the confluence points of the Moskva and Klyazma rivers and included in the dataset. Here, Oka's course winds through a primarily expansive floodplain. Additionally, data of the 230 km of the downstream Volga River (#19) between Volgograd and Astrakhan cities is included in the dataset. Here, the river flows through the steppe zone, and closer to the south its banks are conquered by semi-desert, and is characterized by alluvial sandy channel. Here the dam located at Volgograd significantly influences hydrological regime both by decreasing average and maximal discharges, water levels lowering, and a release wave passage when water level variation immediately downstream the dam can reach 2.5 m day$^{-1}$.

Small rivers located in the Moscow region (#14), including a few rivers within Moscow City (#13) and rivers in the central part of the Nizhny Novgorod region (#18), are also located within the Volga Basin. In the Nizhny Novgorod region, the case study includes the Kudma River and its tributaries – the Ozerka, Setchuga and Pechet' Rivers. The Kudma River is a right tributary of the Volga River downstream of the confluence with the Oka River. The total length of the river is 144 km. The Ozerka River, which is 74 km long, is the largest tributary of the Kudma River. Setchuga and Pechet' are small tributaries of the Kudma and Ozerka Rivers, respectively.

The Setun River (#13) is the largest right tributary of the Moskva River within Moscow City. It flows into the Moskva River at 174 km from its mouth. The total length of the river is 38 km. Eighteen kilometers from its confluence with Moskva, it crosses the Moscow Ring Road and then flows through the city. Its main tributary is the Ramenka River (#13), which is also included in the dataset.

The Ural River (#11) and its principal tributary, the Sakmara (#12), form the border between Europe and Asia and flows through parts of Russia and Kazakhstan. Ural is currently the only major river that flows freely into the Caspian Sea. Channel erosion within the Ural catchment has been the focus of relatively few studies (Yarushina et al. 2009; Sergaliev and Akhmedenov 2014),which propose relatively high values in comparison to nearby rivers, primarily attributed to the river's course through a vast steppe region.

Approximately 4000 km of the primary course of the Ob River (#10) were incorporated into the dataset. The Ob River is situated within the West Siberian Plain, where the geological and geomorphological conditions are relatively uniform. The floodplain and terraces are comprised of alluvial deposits, including sand, sandy clay, and light loam. This results in active riverbank erosion. The bedrock of the Ob River, predominantly found in its lower course, typically comprises solid loamy sediment deposits (Kurakova and Chalov, 2019). The two longest tributaries of the Ob are also part of the dataset. Lower reaches of the Irtysh (#24), a key transboundary river, has been added to the dataset, specifically the section between the Russian-Kazakh border and its mouth. The river's meandering channel has evolved over the past decades under low water conditions, partly due to the regulation of its upper sections by reservoirs. The second longest tributary of the Ob, Chulym (#26), has the lower reach section included, 1100 km upstream from its mouth to the city of Achinsk, where it originates from the low-lying Arga Ridge and extends into the plain, subsequently meandering through a vast floodplain that is flanked by





taiga forests and bogs. Furthermore, more than 220,000 bank erosion sections with total length of 30,200 km of all rivers in
the Ob Basin (#3) and annual discharge values of at least 30 m$^3$/sec were studied.

The adjacent tundra-taiga areas of the low plains are located east of the Ob mouth area. Here, rivers are formed in broad sandy
alluvial deposits composed of marine loam and alluvial sands (Sidorchuk, 2019). With the seasonally thawed layer reaching
about 2 m deep in the area, this area is one of the most severely gullied landscapes in the Russian Arctic, with a gully density
of up to 1–2 km/km$^2$ and a very unstable channel. Three sites of channel monitoring were established in 2022 at the small and
medium Sedayakha, Tyjakha and Khaduta Rivers (#23). The largest river of the Gydan Peninsula, the Messoyakha (#28), was
also included in the base from the confluence with the Nyangus-Yakha River to the head of the delta. It flows under similar
conditions, but has a much higher flow rate, and its channel processes are regulated by the active change of base level due to
the growth of the river delta in the Holocene. Through this process, the rates of horizontal deformations on the river are much
higher than typical for this natural zone.

The low plains' adjacent tundra-taiga regions are situated east of the Ob river mouth area. Rivers originate in expansive sandy
alluvial deposits consisting of marine loam and alluvial sands (Sidorchuk, 2019). With the seasonally thawed layer reaching
about 2 m deep in the area, this area is one of the most severely gullied landscapes in the Russian Arctic, with a gully density
of up to 1–2 km/km$^2$, and rivers here have a very unstable channels. Channel monitoring sites were set up in 2022 at the
Sedayakha, Tyjakha, and Khaduta Rivers, which are small to medium in size (#23). The Messoyakha (#28), which is the largest
river of the Gydan Peninsula, was also included in the dataset from the confluence with the Nyangus-Yakha River to the head
of its delta. Under similar circumstances, it has a much greater flow rate, and the river channel's processes are managed by the
shifting base level resulting from the expansion of the river delta during the Holocene period. This process results in
significantly higher rates of horizontal deformation along the river than would be particularly expected in this natural region.

34 800 km of rivers of the Yenisey Basin (#9) were analyzed, this subdataset consists of 98542 eroded sections of all rivers of
170   the Yenisey Basin that exceed the value of annual discharge of 30 m$^3$/sec. The Yenisei River and its tributaries have mainly
incised relatively straight channels with occasional braided sections and meanders, which are related to the geological structure
of its valley. The banks of this river are composed of massive crystal rocks of Permo-Triassic origin (Saunders et al., 2005).
The delta of the Yenisey River (#21) is 50 km wide and 200 km long. The present study considers over 200 km of the Yenisey
Delta distributary channels.

175   The Selenga River Basin (#20) is the largest tributary of Lake Baikal and belongs hydrographically to the Yenisey River
catchment. The Selenga River is situated between the mountain systems of Southern Siberia (Sayan, Khangai) and the plains
of central Mongolia, draining both southern taiga, forest-steppe, steppe and semi-desert. In the Selenga Basin, there are a wide
variety of geological and geomorphological conditions for the origin of different types of channels, from wide floodplains to
incised ones. The total length of channels of rivers of the Selenga Basin, which was considered in this dataset, is 8,617 km.

180   The Ussuri River (#27) is a major right tributary of the Lower Amur, and for much of its length it straddles the border between
Russia and China. A significant part of the watershed lies in the Sikhote-Alin Mountains, and the basin includes large Lake





Khanka. A 214 km section of the river (with a total length of 897 km) is presented in the database between the Siniy and Wandashan ranges, where it has an actively meandering channel and flows through the broad, swampy Ussuri plain.

34 800 km of rivers of the Lena Basin (#3) were analyzed, this subdataset consists of 132666 eroded sections of all rivers of the Lena Basin that exceed the value of annual discharge of 30 $m^3$/sec. Geological differences between the riverbeds of the Lena Basin influence the channel processes. The upper reaches of the Lena River, together with the Vitim, Olekma, and incised channels, characterize the upper reaches of the Aldan, and bedrock-controlled erosion and stable channels dominate the Vilyuy, while the middle and lower reaches of the Lena River experience active lateral migration, sediment accumulation, and floodplain development. For example, the Lena River near Yakutsk (#1), that is middle course, is mainly anabranching with sandy sediments. Previous studies on channel migration in catchment areas are maintained in works by (Gautier et al., 2021), which suggest significant rates of bank retreat in this region and the key role of permafrost degradation in channel erosion processes. Additionally, the Lena Delta (#8) is the largest distributary channel pattern in the Arctic region, extending 32,000 $km^2$ with 6,000 branches, of which a total 14,600 km is specifically focused on in the dataset. The baseline conditions of channel formation in the Lena River Delta are controlled by river–sea interaction and meteorological factors, as well as past formation features and the continuous presence of permafrost (Chalov et al., 2023a). The largest parts of the ice complex (or yedoma) are located here. Bank erosion estimate covers 90% of the delta terrain, with only 3,300 $km^2$ excluded from the analysis. Four main branches were analyzed for the Lena River Delta: the channels of the Bykovskaya (100 km length), Trofimovskaya (120 km), Tumatskaya (140 km) and Olenekskaya (170 km).

The Yana River (#6) is examined in its upper reaches, covering a 225 km stretch from its origin at the confluence of the Dulgalakh and Sartang Rivers to the confluence with the Adycha River, out of the river's total length of 872 km. This area features a meandering channel that occurs in a broad and deep valley situated within the intermountain depression between the foothills of the Yana Plateau. Permafrost comprises the majority of the eroding banks' length, primarily made up of frozen alluvium, which contains distinct separate ice wedges.

Analysis of the Indigirka River (#16, #5) was conducted employing diverse techniques (both manual and automated) over a 618-km segment spanning from the point where the river enters the lowland, characterised by a sudden decrease in channel slope and a transition in channel type from branching to predominantly meandering, to the beginning of the delta. This area is marked by a broad valley dominated by a large floodplain and the first river terrace, complicated by individual water channels with considerable lengths and relatively low water levels.

For the Kolyma Basin, there is dataset (#7) that provides estimates for the 300 km downstream sections of the river system and its estuary. In this area, the Kolyma has a mainly meandering sandy channel with significant outcrops of yedoma sediments with high ice content (Strauss et al., 2021). Sparse evidence suggests (Murton et al., 2015) that channel dynamics have increased over recent years due to the climate-driven collapse of yedoma outcrops. In addition, 8100 km with 113875 eroded sections of all rivers of the Kolyma Basin (#4) that exceed the value of the annual discharge of 30 $m^3$/sec were analyzed. Significantly smaller than the Lena Delta, the Kolyma River Delta (#22) consists of two main branches and is also formed by the accumulation of sediments from the river and from the erosion and abrasion of ancient landforms due to sea-level





fluctuations and tectonic movements of the Earth's crust. The total length of the estimated channel distributary network in the Kolyma Delta is 120 km, covering over 3,200 km² of the delta.

The Kamchatka River (#15), located at the easternmost point of the dataset, is the largest river of the Kamchatka peninsula situated in the Russian Far East. The 580 km section that was analysed spans approximately 77% of the total river length,
commencing where the river exits the mountains and enters the Central Kamchatka plain, and terminating in a river gorge within the Kumroch range, just before the river's mouth area begins.

**2.2 Bank erosion dataset and methods**

Owing to differences in river sizes and the limitation of the analyses to particular river reaches, the compiled dataset relies on five main tools applied at various spatial levels (Table 2). Compatibility and validation of the tools are discussed in the next
section. All methods provided data on bank erosion rates (m/year), whereas area-based approaches provided additional data on the area (m²/year) and volume (m³/year) of bank collapse due to erosion (and mass of sediment release, t/year):

- method I: on-site local measurements based on field-based measurements of bank erosion in a particular reach;
- methods II, III, IV: linear methods, which are applied for extended river reaches based on tools that track either the banklines (obtained by manual digitizing [method II] or by the application of a specific GIS digitizing algorithm
[method IV]) or stream centerlines (method III);
- method V: area-based approaches, which are based on classifications of water and land from satellite images and further applied to quantify riverbank erosion as transitional pixels from land to the river.





**Table 2. Overview of applied tools**

| Method | I | II | III | IV | Va | Vb |
|---|---|---|---|---|---|---|
| Applied tools | Manual measurements with geodesic instruments and UAVs | Manual digitizing in ArcGIS | Centerline method in ArcGIS | Digital Shoreline Analysis System (DSAS) | Rstudio, ArcGIS | |
| | | | | | using existing databases | using water indices |
| Site numbers | 14, 18, 23 | 1, 7, 10, 15, 16, 19, 28 | 11, 13,12, 20, 24 | 17 | 2, 3, 4, 9 | 5, 6, 8, 21, 22, 25, 26, 27 |
| Rivers | Moscow region rivers, Kudma catchment rivers, Yamal region rivers | Lena; Kolyma; Ob, Indigirka, Volga, Kamchatka, Messoyakha, | Ural, Sakmara, Setun and Ramenka, Selenga catchment, Irtysh | Vistula | Lena, Ob, Yenisey, Kolyma basins | Yenisey, Lena, Kolyma deltas; Indigirka, Oka, Chulym, Ussuri, Yana |
| List of parameters in dataset | Bmean, Bmax | Bmean, Bmax, Amean | Bmean | Bmean | Bmean, Amean, Bmax, Vmean | |


**I. On-site local measurements.** Measurements taken directly on-site rely on regular surveys of a river's planform, a method suitable for monitoring deformation in small rivers (#14,18). Rods are positioned along the banks, especially those that have eroded, at a certain distance from them to create a grid that spans the area to be mapped. Two tapes were utilised to record data along the transect: one measured the distance between two rods, while the second tape measured the perpendicular distance

from the first tape to specific points on the eroded bank (e.g. the top, toe, etc.) at intervals between the rods. The site's bank erosion intensity data could be obtained with a level of accuracy of 1 cm. Orthophotography and a digital elevation model (DEM) surface were also generated annually via UAV-based surveys during field campaigns (Fig. 2), covering the Yamal Peninsula rivers (#23). Estimates of some rivers within Moscow city (such as the Setun, Ramenka, #13) - particularly unaltered and non-channelized sections - were combined with satellite imagery processing. A combination of aerial images from 1942

and Google Earth images from 2010 was made.





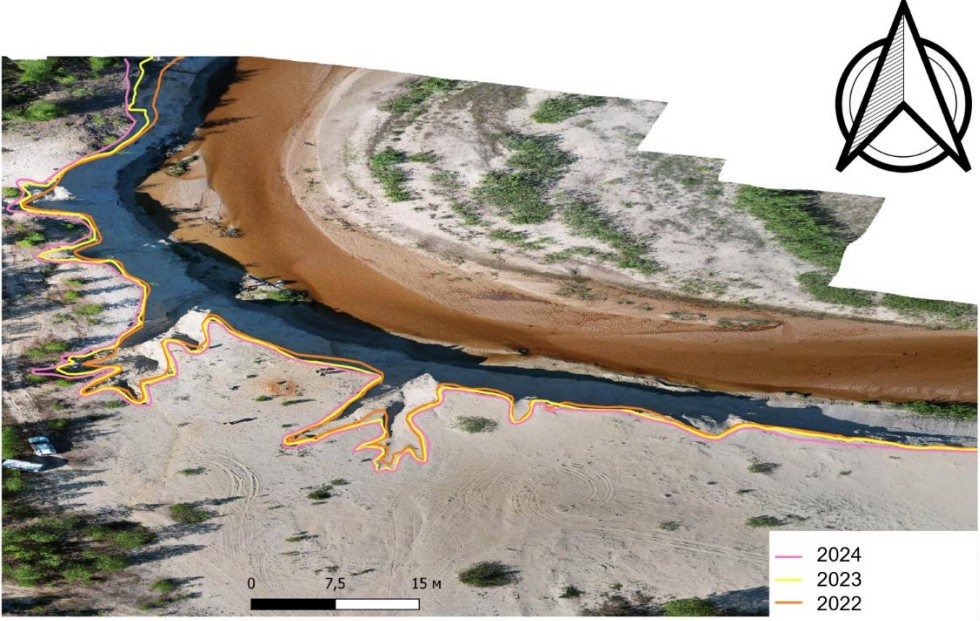

**Fig. 2. Location of bankline revealed from UAV at the Tayakha River at Yamal (orange – September 2022; green – August 2023; red – July 2024)**

**II. Manual digitizing.** This method of riverbank migration analyses was undertaken by comparing satellite images captured
at times of similar water flow rates (low-water periods in August or September) with a deviation of 5% from standard discharge
levels. Satellite images from the CORONA series, which have a resolution of 1.8 m, and those from the Landsat 5 satellite
with a 30 m resolution, were chosen as the earliest available images. The current position of the riverbank line was evaluated
using multispectral satellite images obtained from Sentinel-2 with a resolution of 10 m and Landsat 7 and 8 with resolutions
ranging from 15 to 30 m. The methodology involved comparing the positions of riverbank lines that had been digitized from
satellite images taken in various years, as shown in Figure 3. Consequently, we identified riverbank erosion fronts, and used
ArcGIS tools to determine the mean and maximum annual erosion rates, as well as the area of erosion. Riverside bank retreats
were also calculated using manual image analysis of Keyhole imagery dating back to around 1964-75 and subsequent detailed
analysis of Quickbird, Worldview, and Spot satellite images from approximately 2012-18.





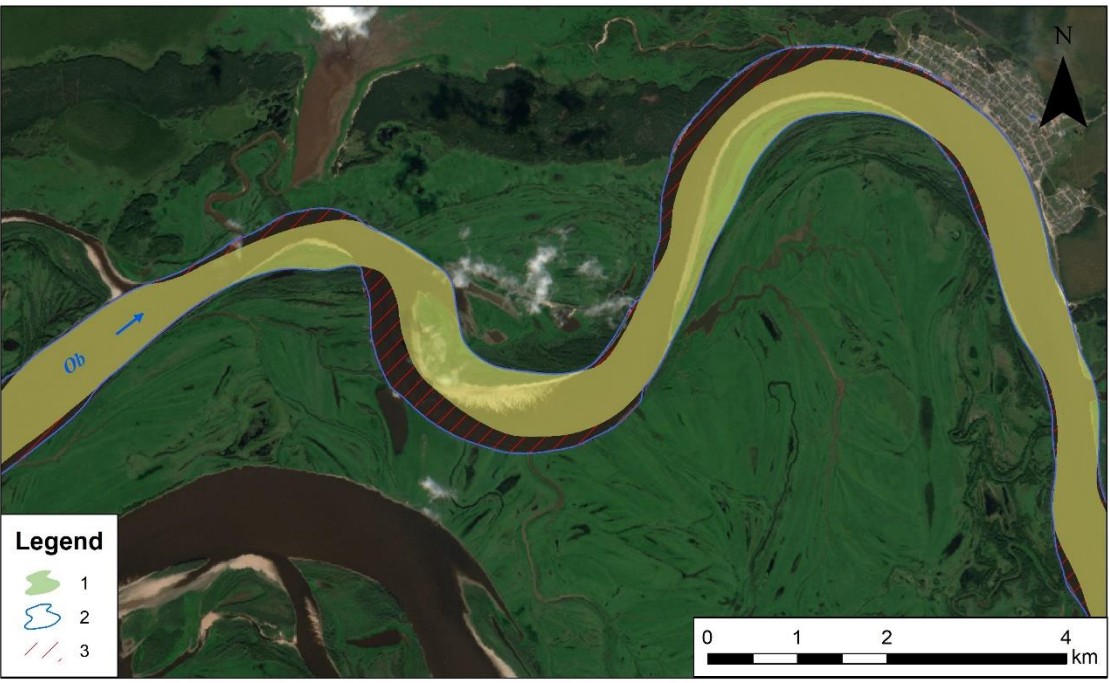

**Fig. 3. Comparison of positions of riverbank lines of Ob (#10) middle reach : 1 – 1970 year, 2 – 2018 year, 3 – channel erosion areas (1970–2018) (© Esri Imagery)**

**III. Tracking river centerline movement.** The method of long-term channel migration was based on river centerline calculation from satellite images in ArcGIS. The methodology was based on the classification of satellite imagery using the *Maximum Likelihood Method,* which involves identifying and digitizing three training sets: a) water surface or fluvial bed; b)
island banks, bars, areas without vegetation and sand; c) islands covered by perennial vegetation. These training sets are used to generate a classified raster output for each satellite image. The water surface was automatically extracted from a combination of visible red, NIR, and SWIR channels, and erosion polygons were calculated as the positive difference between the 2019 and 2000 water surface rasters. For each erosion polygon, the centerline was constructed and then smoothed using a Savitzky-Golay linear filter, which allowed the construction of near-parallel transects that were used to parameterize the erosion rate for
points every 10 m. This approach was implemented along 8,617 km of rivers in the Selenga Basin (#20), Irtysh (#24), Ural (#11), and Sakmara (#12) rivers.

**IV. Digital Shoreline Analysis System (DSAS) in ArcGIS.** For some of the case studies, the application of the Digital Shoreline Analysis System (DSAS) was tested based on an automated statistical model deployed to estimate riverbank erosion/accretion along a selected reach of the Vistula River's middle stream. The DSAS model developed by USGS as a key
component of its "Coastal Change Hazards" program calculates a comprehensive array of regression statistics within a systematic, readily repeatable method that can be implemented on a large amount of data (USGS 2019). The DSAS model is operating based on statistical estimation methods to calculate the rate-of-change statistics from satellite data. DSAS calculates erosion and accretion based on a time series of vectorized shoreline positions marked by transects generated from a referenced

baseline (Himmelstoss et al. 2021). In this study, bankline assessment was carried out using two different measurement

calculations: the end-point rate (EPR) and the shoreline change envelope (SCE). The EPR is calculated by dividing the distance

between two given shorelines by the time elapsed between the oldest and most recent shoreline, whereas SCE reports a distance

(in meters) and does not document a change rate. In this analysis, the DSAS model was employed in the selected study area,

utilizing distinct Landsat images of the Vistula River spanning between 2006 and 2023 to quantify riverbank erosion and

accretion by implementing the following steps (Fig. 4). Thus, the persistent bank line of distinct years (2006 and 2023) was

delineated using Landsat images to assess erosion/accretion via DSAS. The generated binary raster datasets were then

transformed into vector data, and the land–water boundary was demarcated. DSAS needs a single land–water boundary as

input, which is called the "baseline", and the erosion/accretion is calculated relative to the baseline. The baselines created for

both the right and left banks had a buffer of 200 m, and the transects were cast perpendicular to the baseline at intervals of 100

m (Fig. 4).

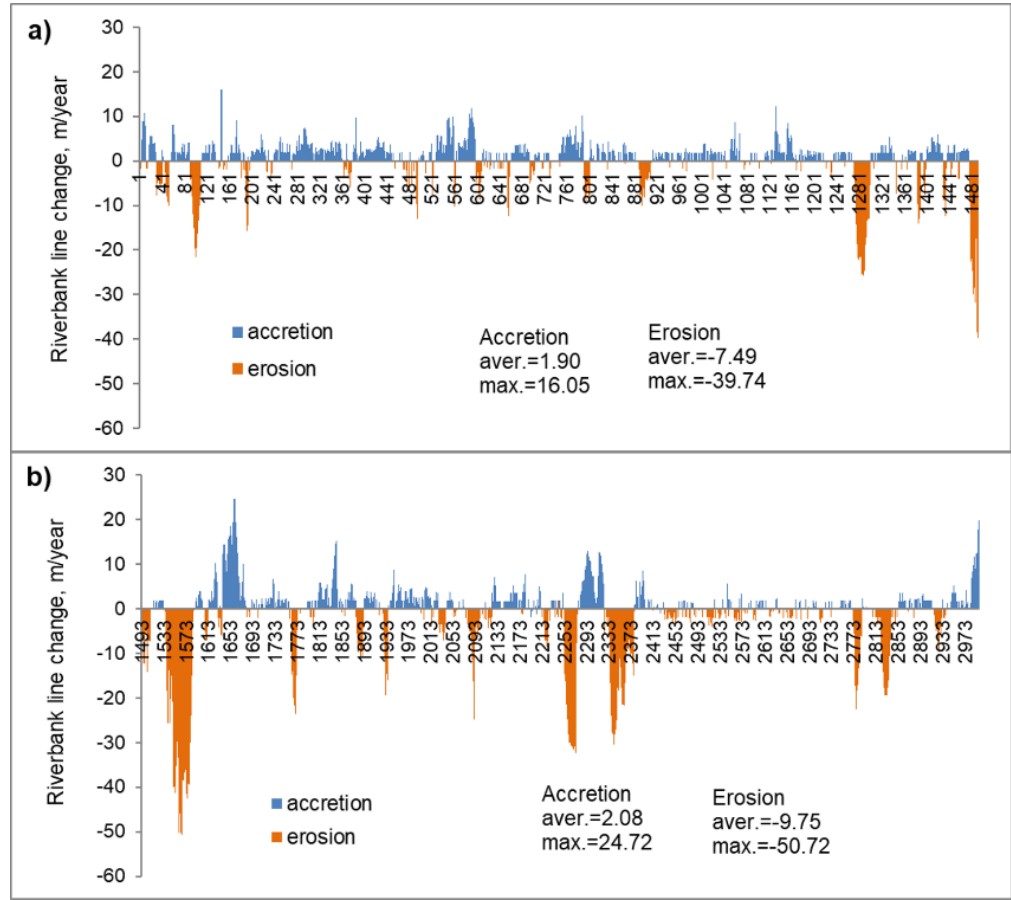


**Fig. 4. Endpoint Rates (EPR) calculation curve generated from DSAS for riverbank line change (m/year) of the Vistula River (#17) in the middle section between the mouth of the Radomka River (km 430) and of the Narew River (km 551): A – left bank, B – right bank. X-axis shows the transect number. The numbering of the transects increases from south to north.**



**V. Area-based approach for rivers**. The area-based method was used to assess river planform changes was used similar to
the SCREAM method (Rowland et al., 2016) and REAL dataset (Langhorst and Pavelsky, 2023). Channel erosion was
measured as a bank retreat along the studied rivers based on the Global Surface Water Explorer (GSWE hereinafter) automatic
image interpretation dataset, which provides global data on the location and persistence of surface water and its changes from
1984 to 2021. Each year 14–22 space images (raster layers) that characterize periods of different water flow. In our research,
the "Water Transitions" layer was used, which provides an estimate of long-term water history by identifying transitions
between permanent water, seasonal water and land classes between the first and last years. Water surface area changes on the
images are mainly caused by water balance fluctuations; however, in the regional context of river channels, channel erosion
can be described as a new permanent water body adjacent to the modern river channel.

Another GSWE layer, "Occurrence Change Intensity", provides information on where the surface water occurrence increased,
decreased or remained the same for 1984–1999 and 2000–2021, whereas the "Transitions" layer divides these changes into
eight classes based on seasonality and stability. Both classifications were compatible and were used similarly (Fig. 5). To
obtain the migration rate [m/yr] for each reach expressed, the right- and left-bank migration polygons were divided by the total
surface, then by the length of the oldest of the two channel banks and then by the number of years of the analyzed time interval.

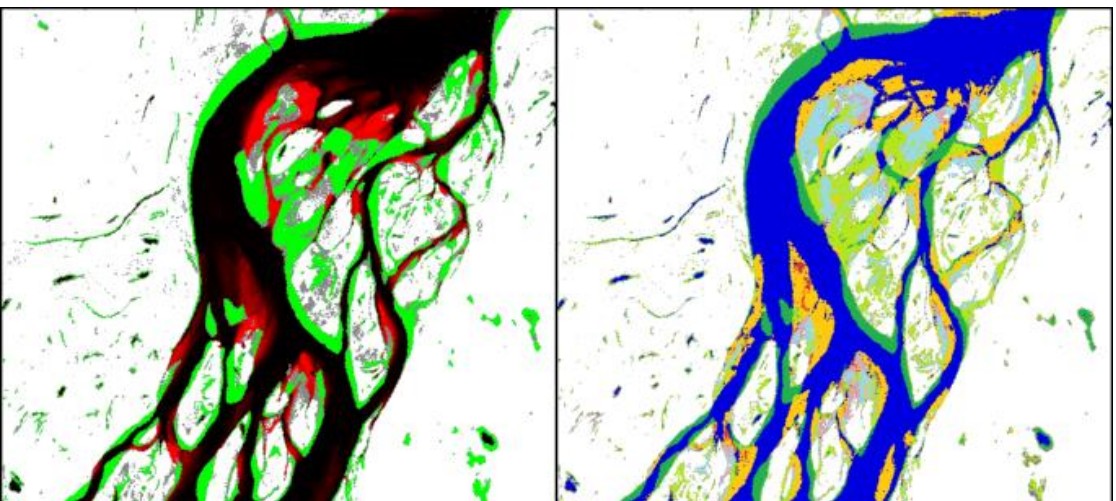

**Fig. 5. Bank erosion classification according to GSWE map – change intensity (left) and transitions (right) (Lena River case study)**

Also, manual area-based approach for distributary channels was conducted by comparing two Landsat satellite images with
similar water flow conditions for deltas of the Yenisey, Lena and Kolyma Rivers and long meandering stretches of Indigirka,
Oka, Chulym, Ussuri and Yana rivers in homogeneous landscape and flow conditions. Bank locations were identified using
the AWEI water index, which made it possible to clearly distinguish the boundary between water and land and highlight hidden
details that are poorly visible when using only visible channels, including submerged shallows and shadowed sections (Feyisa
et al., 2014).





## 2.3. Compatibility of different methods applications and uncertainty estimate

The construction of the dataset involved a combination of manual and semi-automatic methods for estimating bank erosion, incorporating both field-based and remote-sensing data; consequently, multiple sources of uncertainty could affect the results. The dataset primarily showcases the potential of employing various methods to measure channel displacement rates and

qualitatively comparing them across territories with diverse natural conditions and spatial factors driving erosion. Considering the scale of the observed phenomena, it's worthwhile to examine the extent of bank retreat extremes. In studies of bank erosion, the maximum values of bank retreat often receive the most attention, however, during the process of automatically defining a bank's edge, an increase in retreat distance typically results in a corresponding increase in error values or the likelihood of bank position error. Consequently, the results of automatic delineation necessitate thorough verification in this analysis due to

disparities in bank failure patterns and the bank slope's configuration and incline.

Manual digitizing of high-resolution satellite or aerial imagery is the most accurate technique for bank retreat calculations, yet it is also the most labor-intensive due to a time-consuming process and limited accessibility compared to Landsat and Sentinel databases (Piégay et al., 2020). The availability of archived satellite and aerial imagery enables a substantial expansion of the comparison. The accuracy of manual delineation is influenced by the reference scale used during the process, which is often

referred to as the "eye altitude," and typically remains below 1-2 meters of error. The primary causes of manual digitization mistakes are tied to the precision of georeferencing (both the primary and secondary images) and variations in tilt angle. A case study of the Mekong Delta (Binh et al., 2020) found that the total digitization error was no more than 2.8 metres.m. Up to this point, numerous methods have been devised for categorizing optical images and determining landscape boundaries from them, resulting in comparable levels of accuracy to manual digitization, specifically 0.4 to 12.7% for the erosion area in the

Colville River case study (Payne et al., 2018).

The accuracy of our riverbank migration estimates was assessed by determining the bank erosion rate for several river stretches using multiple approaches. The middle course (#1) of the Lena River included several reaches that were observed on islands and on both the right and left banks, as shown in Table 3. Images captured by Landsat 5 in August 1999 and Landsat 7 in August 2020, both taken when water levels were low, were manually digitized to verify the GSWE results for the corresponding

timeframe. The error was determined by comparing it to the percentage discrepancy between manual tracing and erosion detection. The findings suggest a compatibility range of 0.4 to 4.1%, thereby supporting the idea of compatibility among the tools used in this study.

**Table 3. Validation of area-based calculations**

| Reach length, km | Areas of bank erosion A, $m^2$ | | Error, % |
|---|---|---|---|
| | Manual digitizing | GSWE | |
| 5 | 1,646,531 | 1,612,004 | 2.1 |
| 4 | 1,038,047 | 999,325 | 3.7 |



| 3 | 354,990 | 353,679 | 0.4 |
|---|---------|---------|-----|
| 3 | 652,891 | 648,843 | 0.6 |
| 8 | 2,396,625 | 2,297,342 | 4.1 |

Previous studies (Albertini et al., 2022; Huang et al., 2018; Laonamsai et al., 2023; Liu et al., 2022; Zhou et al., 2017) have evaluated the precision of different water surface definition indices in identifying the precise location of riverbanks, with results showing that indices like MNDWI and AWEI yield an average error rate of up to 5%. The results indicated that automatic computations can be applied when the rate of streamflow surpasses 100 m$^3$/s, thus enabling the calculation of outcomes with an accuracy of under 10%. The accuracy of optical extraction is constrained by various morphological features

of the banks, such as sandbars and debris resulting from bank material collapse near the edge, the shadows cast by the banks and trees, and fallen tree trunks. A similar investigation was carried out for the portion of the Yana River that was examined, employing multispectral (Landsat 8) and optical (WorldView-2) images acquired on 21/06/2021, as shown in Figure 6. Banks were automatically classified using unsupervised classification and the AWEI index in Landsat imagery. For the comparison, the images were also digitized at a scale of 1:500 using a high-resolution photograph. The research shows that automated

methods for marking bank positions result in minimal average errors when dealing with uniformly well-lit slopes and the water-dry land boundary, but the error rate increases for other conditions. At the same time, the average error for the 6 assessed locations (after smoothing the bank line using the Savitzky-Golay method) was 7.5 meters for automatic classification and 3 meters for the AWEI index. Annual retreat rates combined with prolonged periods of comparison will lead to extremely low total percentages, specifically less than one percent.

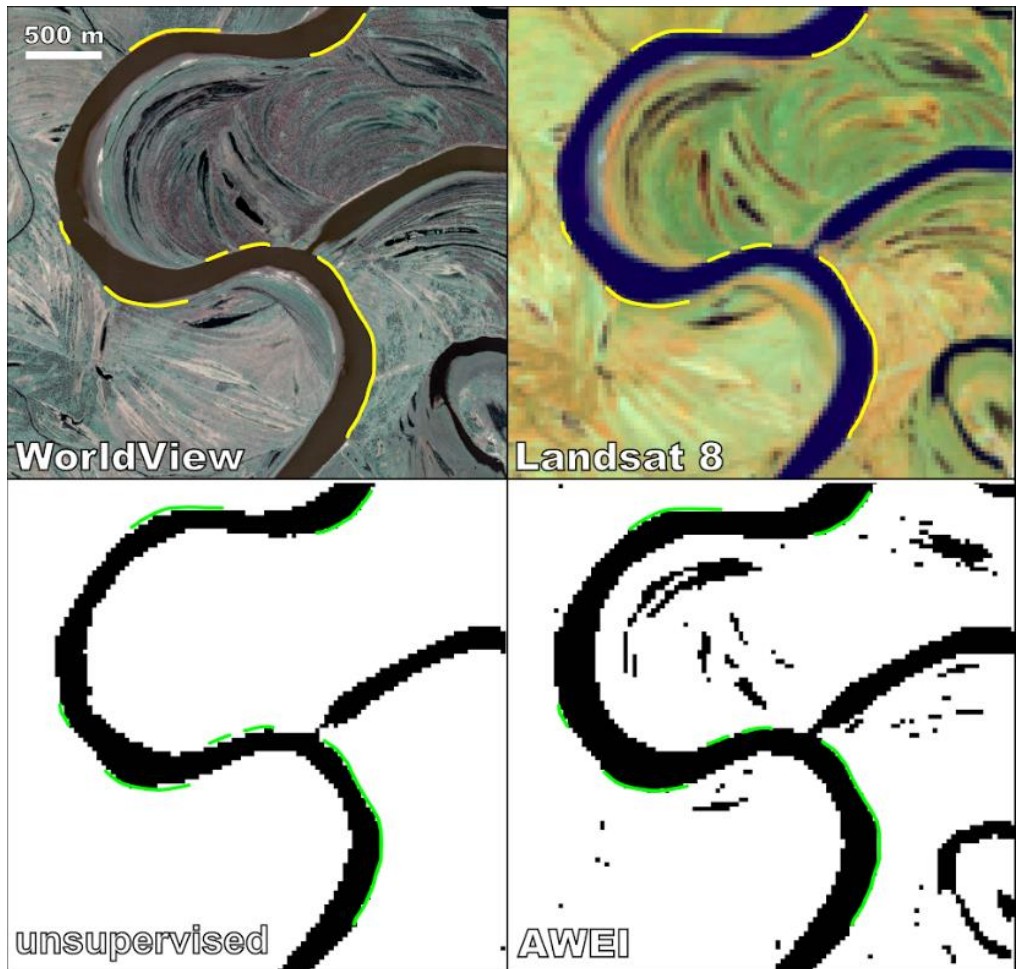


**Fig. 6. Yana River section on satellite images from June 21, 2021 with digitized position of the riverbanks in the erosion area (top) and an example of their automatic detection by automatic classification (bottom left) and with application of the AWEI index (bottom right).**

Uncertainty can also be attributed to fluctuations in river water levels. To minimize this source of error, discharge estimates

for all rivers were utilised for the image dates in question, ensuring conditions with a discrepancy of less than 5%. The greatest

inaccuracies are typically seen when employing the centerline migration approach, primarily due to variations in flow rate.

The placement of centerlines is influenced not only by the local characteristics of the erosion site but also by the entire width

of the channel, making low-water images unsuitable for this approach, and it is essential to select an image taken on a date

with discharges similar to those at bankfull conditions. The error was estimated by comparing the results of manual digitization

for different sections of the Irtysh River, using pairs of images taken on different dates, with one pair corresponding to the

same stage of the water regime and another pair to similar discharge levels. The analysis revealed discrepancies of up to 7.4%

in average annual retreat rates and up to 6.3% in maximum rates when comparing yearly rates calculated from different image

pairs (Table 3).





**Table 3. Validation of erosion-rate-based calculations**

| Bank erosion rates, m/year (mean/max) | | Error, % (mean/max) |
|---|---|---|
| Same phase of the water regime (Aug 1987 – Aug 2020) | Similar water flow conditions (13/09/1987, 2040 m$^3$s$^{-1}$ – 25/09/2020, 1990 m$^3$s$^{-1}$ | |
| 7.6 /12.6 | 8.0 / 12.2 | 5.3 / 3.3 |
| 6.8 / 12 | 7.3 / 12.8 | 7.4 / 6.6 |
| 10.9 / 16.4 | 11.4 / 17.2 | 4.6 / 4.9 |
| 9.5 / 15.6 | 9.7 / 15.6 | 2.1 / - |
| 9.0 / 14.4 | 9.2 / 15.3 | 2.2 / 6.3 |
| 5.9 / 9.6 | 5.8 / 9.6 | 1.7 / - |


 We compared retreat rates from key river sections using both semi-automatic and manual methods across various time periods to evaluate the hypothesis that retreat characteristics become more consistent with longer comparison intervals. The data collected at the key sites on the rivers within the Ob basin (#10) were found to be similar when obtained using both semi-automatic and manual methods, and allowed for results with an acceptable error margin of up to 10%. Statistical analysis of
the 260-kilometer Indigirka river section, which was digitized both manually (using high-resolution images from KeyHole and WorldView over a 42-year period from 1975 to 2017) and semi-automatically (using Landsat imagery over a 19-year period from 2000 to 2019), revealed that at cross-sections with a 30-meter frequency, the error in the mean retreat rate is approximately zero for half of the sections with either small or extreme retreat rates, even when comparing different periods (as shown in Figure 7). Automated retreat rates for the longer time period are overestimated by approximately 6.5% compared to manually
calculated ones. Based on the above comparisons, we conclude that the retreat rates, aggregated in the presented database, are comparable to each other despite using different spatial approaches and time scales, and the total error of the obtained rates does not exceed 10%.

Earth System
Science
Data

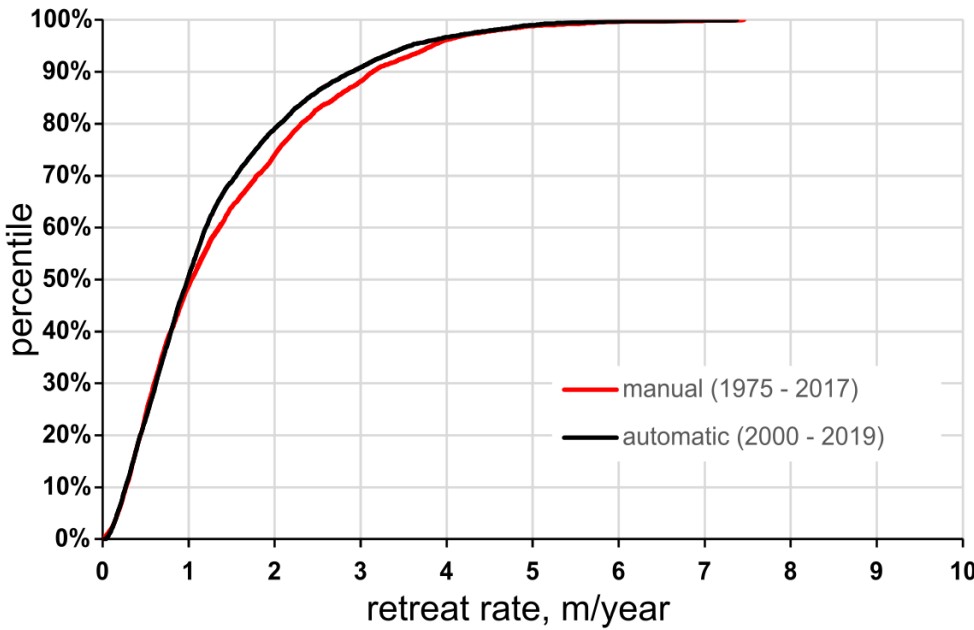

**Fig. 7. Percentile curves of the retreat rates for manual and semi-automatic delineation of banks on the similar section of the Indigirka River.**

DSAS approach (methods IV) was verified by comparing measurements from three control cross-sections using geodetic measurements with a GNSS RTK receiver for the Wistula River. The 2006 data came from measurement cross-sections provided by State Water Holding Polish—Water Management Authority in Poland. The validation results show that DSAS signifies its utility and reliability—the correlation coefficients ranged from 0.95 to 0.99. For example, study by (de Lima et al., 2021), the in-built EPR4Q for QGIS was validated with DSAS, depicting a high correlation coefficient range of 0.98 - 1.00 across various types of shorelines, thereby affirming the accuracy of DSAS in shoreline change assessment. (Gómez-Pazo et al., 2021), the ODSAS ("Open Digital Shoreline Analysis System"), was developed and it was compared with the DSAS, which yielded similar results. These studies highlight the credibility of DSAS in bankline and shoreline assessments.

**2.4. Sediment yield estimates**

Volume assessments of bank erosion used in the paper using formula (1), which involves determining the eroded area through manual satellite data processing or automatic image classification data from GSWE.

$$W_{ch} = \frac{S_{er} \cdot \rho_{sed} \cdot (h_b + h_d)}{\Delta t}, \tag{1}$$

With $\rho_{sed}$ – riverbank sediments density [kg/m$^3$], $\Delta t$ – time gap between satellite images (y), $S_{er}$ – eroded area from satellite data [m$^2$]; $h_b$ – bank height [m]; $h_d$ – river depth [m].

The bank height (above average iced low-flow period water level $h_b$) was obtained from Arctic DEM digital elevation model with a resolution of 2 m (Morin et al., 2016). The bank height calculation was made using programming in the *R* language (*terra* and *sf* packages). For each group of adjusted pixels of channel erosion area $S_{er}$, a buffer zone of 50 meters radius was



created from which the maximum and mean value of absolute height was calculated using Arctic DEM data. To exclude the influence of canopy, buildings and other outlier errors, values of 0.95 and 0.05 quantiles was used. The difference between
maximum (0.95 Q) and minimum (0.05 Q) value of elevation of each buffer zone can be described as the difference of the low-flow period level of the river and the present bank and floodplain height.

The underwater part of bank slope was obtained as a mean depth of river using the 1D Shezi formula (2–4) based on data obtained from global datasets HYDROAtlas (Linke et al., 2019) and GRWL (Allen and Pavelsky, 2018) under average annual flow conditions. Data of mean annual discharge and water slope were obtained from HYDROAtlas with mean resolution of
~4 km. Data of mean river width was obtained from GRWL with mean resolution ~3 km. The Manning's roughness coefficient is assumed as constant for all rivers as 0.045 (Baryshnikov, 1990).

$$Q = \omega \cdot C \cdot \sqrt{RI}, \qquad (2)$$

$$C = \frac{1}{n} h^{\frac{1}{6}}, \qquad (3)$$

$$h_d = \left(\frac{Q \cdot n}{B \cdot \sqrt{I}}\right)^{\frac{3}{5}}, \qquad (4)$$

with $Q$ – water discharge [m³ sec⁻¹]; $\omega$ – cross-section square [m²]; $C$ – Shezi coefficient [m⁰·⁵ sec⁻¹]; $R$ – hydraulic radius [m]; that is similar as river width $B$; $I$ – slope; $n$ – roughness coefficient.

An evaluation of the bulk density of riverbank sediments was comprised as for typical rivers sediment values from (Karaushev, 1977) and included into the dataset as additional parameterization (see 2.5). Also, the height of the eroded bank edges was obtained from the ArcticDEM elevation model. Bank edge height is calculated automatically using the Extreme Difference
Estimator method in the $R$ environment that searches for graph inflections (corresponding to the seam and the edge of the coastal slope) and records the difference in the heights of the inflection points in each transect. The output is erosion sections, formalized into points located along their central line with the required density. Each point contains all the required information for further processing – erosion width, erosion wall height, and distance along the bank line.

## 2.5. River reach classifications

Additionally, each river section was characterized by natural drivers of channel evolution. The annual water discharge and river characteristics were taken from the HydroATLAS database (Linke et al., 2019) with a spatial resolution of 10 km. Attribution of river to natural zones was done based on the Köppen climate classification (Peel et al., 2007), which is indexed based on three letters of the classification scheme (e.g., BSk relates to a dry [B], semi-arid [S], cold [K] climate).

Permafrost zones were categorized based on actual area underlain by permafrost from Obu et al. (2019). Finally, rivers were
classified by channel patterns distribution according to the map: "Channel morphology regime of rivers of USSR" (Chalov et al., 2018) and further classified by dominant bed-deposit types. Each bed deposit class (from sand to gravel) was characterized by specific values of grain density (see Table 4), which is used to estimate volumes and masses of sediment delivery to river channels.

**Table 4. River classifications used in the dataset**



| Parameter | Classes | | | Description | Source |
|---|---|---|---|---|---|
| Water runoff | (m³/s) | | | Unique values for each bank retreat site with 10 km averaging | HydroATLAS (Linke et al., 2019) |
| Permafrost | Cont | | | Continuous permafrost domain 90%> | (Obu et al., 2019) |
| | Discon | | | Discontinuous permafrost domain 50–90% | |
| | Spora | | | Sporadic permafrost domain 10–50% | |
| | Isol | | | Isolated permafrost domain 0–10% | |
| Natural zone | **A** Tropical | **f** Rainforest **m** Monsoon **w** Savanna, dry winter **s** Savanna, dry summer | | The Köppen climate classification | (Peel et al., 2007) |
| | **B** Dry | **W** Arid desert **S** Semi-arid steppe | **h** Hot **k** Cold | | |
| | **C** Temperate | **w** Dry winter **f** No dry season **s** Dry summer | **a** Hot summer **b** Warm summer **c** Cold summer | | |
| | **D** Continental | **w** Dry winter **f** No dry season **s** Dry summer | **a** Hot summer **b** Warm summer **c** Cold summer **d** Very cold winter | | |
| | **E** Polar | **T** Tundra **F** Ice cap | | | |
| Geology | 209 kg/m³ – Yedoma outcrop | | | Mean value of sediment density with an ice content assessment by (Kizyakov et al., | Map: "Channel morphology regime of rivers of USSR" |





| | 2024) (Fuchs et al., 2020) | |
|---|---|---|
| 1,100 kg/m³ – Sand with silt river sediments<br>1,500 kg/m³ – Sandy sediments<br>1,700 kg/m³ – Sand with gravel sediments<br>2,100 – Gravel boulder sediments | Sediment density based on mean sediment diameter by (Karaushev, 1977) | |

**2.6. Online platform**


The multi-tool dataset on Large Northern Eurasian Riverbank migration (NERM) is realized via planform GISCARTA, which provides online access to the dataset, its visualization, and download of data as GEOJSON and text files of attributive tables (Fig. 8). All features of the database, polygons of bank retreat, points of maximum values of bank erosion centrelines, have their attribute parameters each in a separate column. There are values of bank retreat rates in meters per year (Bmean), channel

erosion area in square meters per year (Amean), mass of channel erosion in tons per year (Vmean), maximum values of bank retreat (Bmax), start and end years of satellite images (year_start, year_end), name of the river (river), mean annual water discharge (m³/sec). The platform is available via the link ([https://map.giscarta.com/viewer/93a6a4b3-179f-450f-be02-a31ca6db245b](https://map.giscarta.com/viewer/93a6a4b3-179f-450f-be02-a31ca6db245b)). The dataset is constantly updated and includes broader results than those discussed in the paper. In addition, it is possible to get access to the data in Zenodo, where ESRI shape files are stored (Chalov, S., Ivanov, V., Danila, S.,

Pavlyukevich, E., Habel, M., Botavin, D., Chalova, A., Golovlev, P., Kamyshev, A., Kolesnikov, R., Koneva, U., Kurakova, A., Mikhailova, N., Tuzova, E., Prokopeva, K., Zavadsky, A., Acharyya, R., Chalov, R., Varenov, A., 2024). Using large datasets and considering evolutionary trends, aggregating data via boxplots provided an effective method for synthesizing results and addressing key discussion points.





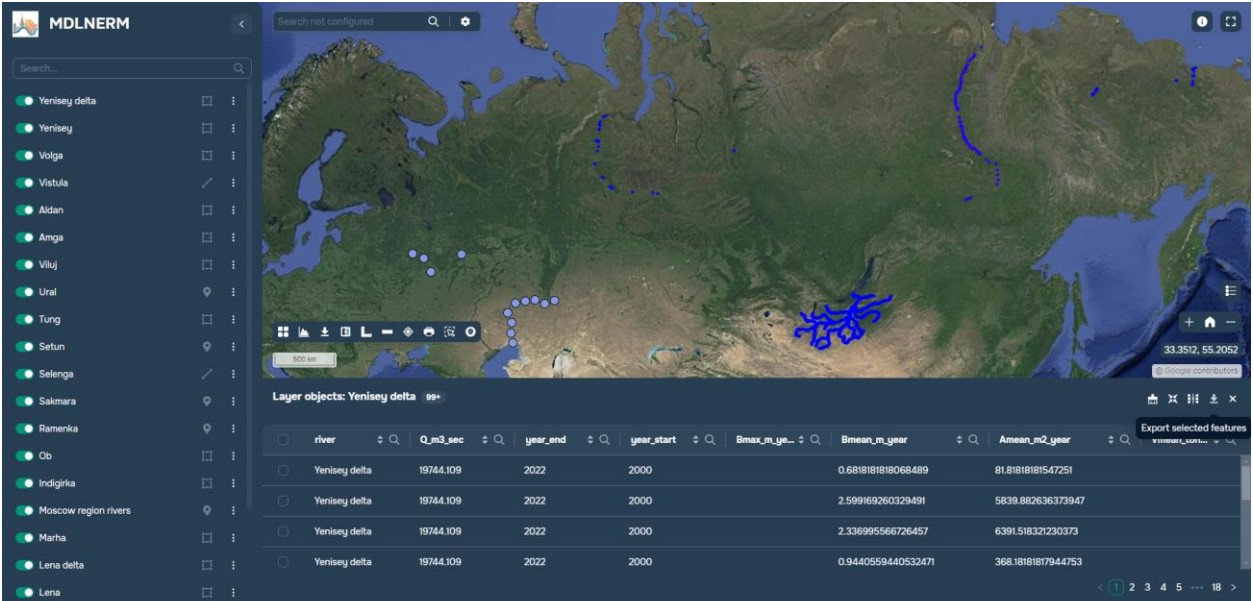


**Fig. 8. GISCARTA platform screenshot**

## 3 Results

In total, this dataset consists of a multi-tool dataset of channel erosion rates for 257 rivers and three deltas with annual mean water runoff data from less than 0.2 m$^3$/s in small rivers of Moscow city region to 19,700 m$^3$/s in the Yenisey downstream. It

covers 626 772 reaches with defined bank retreat rates for the 20–30 years and more. The spatial resolution of the reaches in the dataset is from 10 m to 2 km.

Values of mean bank erosion rates for whole rivers ranges from 0.01 to 53 m/year. Only for the manual data from Kudma catchment (18) is there no calculation of this parameter. Mean value of this series is 1.38 m/year, median values is 0.83 m/year. The quartile of 75% has the value of 1.62 m/year, and for the 25% quartile this value is 0.54 m/year. The distribution of the

observed riverbank migration rate is approximately gamma (Fig. 9).

Much smaller rivers contain parameters of maximum bank retreat (m/year) that can be detected through methods I and II. Within 21 rivers processed by these approach ($n = 2859$), values of maximum bank retreat for whole rivers are between 0.01 and 26.3 m/year with a mean value of 2.53 m/year. The quartile of 75% has a value of 3.39 m/year, and its 25% is 0.88 m/year. The distribution is log-normal.





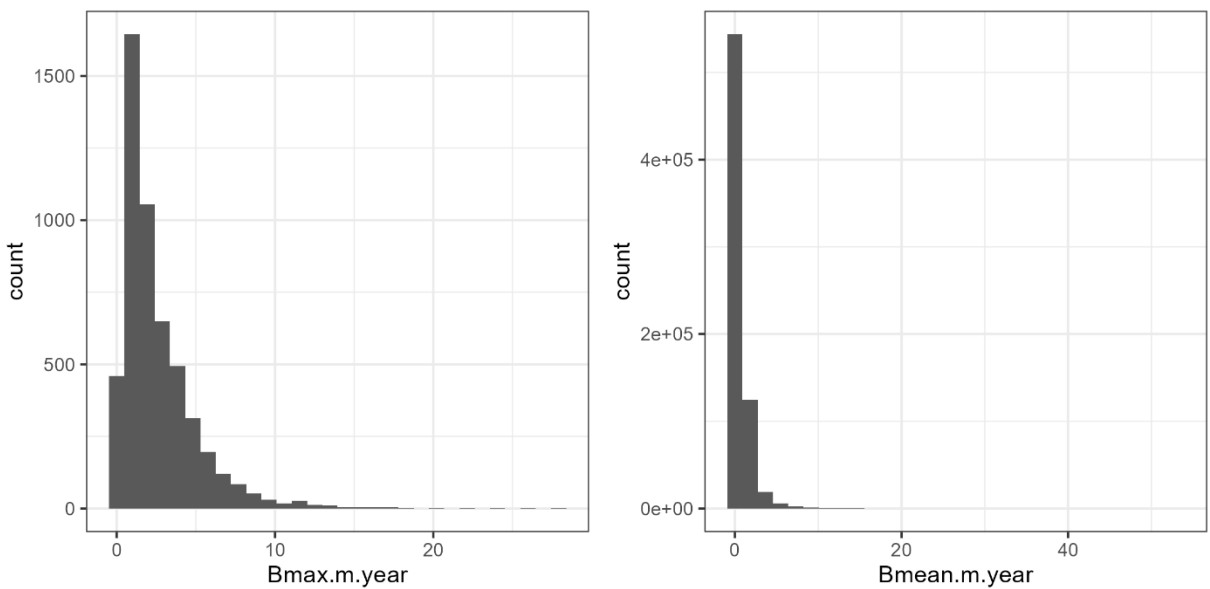


**Fig. 9. Distribution of observed riverbank erosion rates in Northern Eurasia (left – annual bank retreat B [m/year]; right – maximal bank retreat B [m/year])**

To describe this dataset, the boxplot method was used to describe Bmean, Bmax and Amean (Fig. 10, 12, 15). Boxplot content here are "minimum" value (Q1 - IQR), first quartile [Q1], median – line, mean – cross, third quartile [Q3] and "maximum",

(Q1 + IQR), outliers – point) – of sediment flux rate (Mt/year/km). IQR is the interquartile range (IQR) or the 50 percent of data points lying above the first quartile and below the third quartile. The largest values of mean bank erosion are for the Vistula and Volga Rivers (Fig. 10).

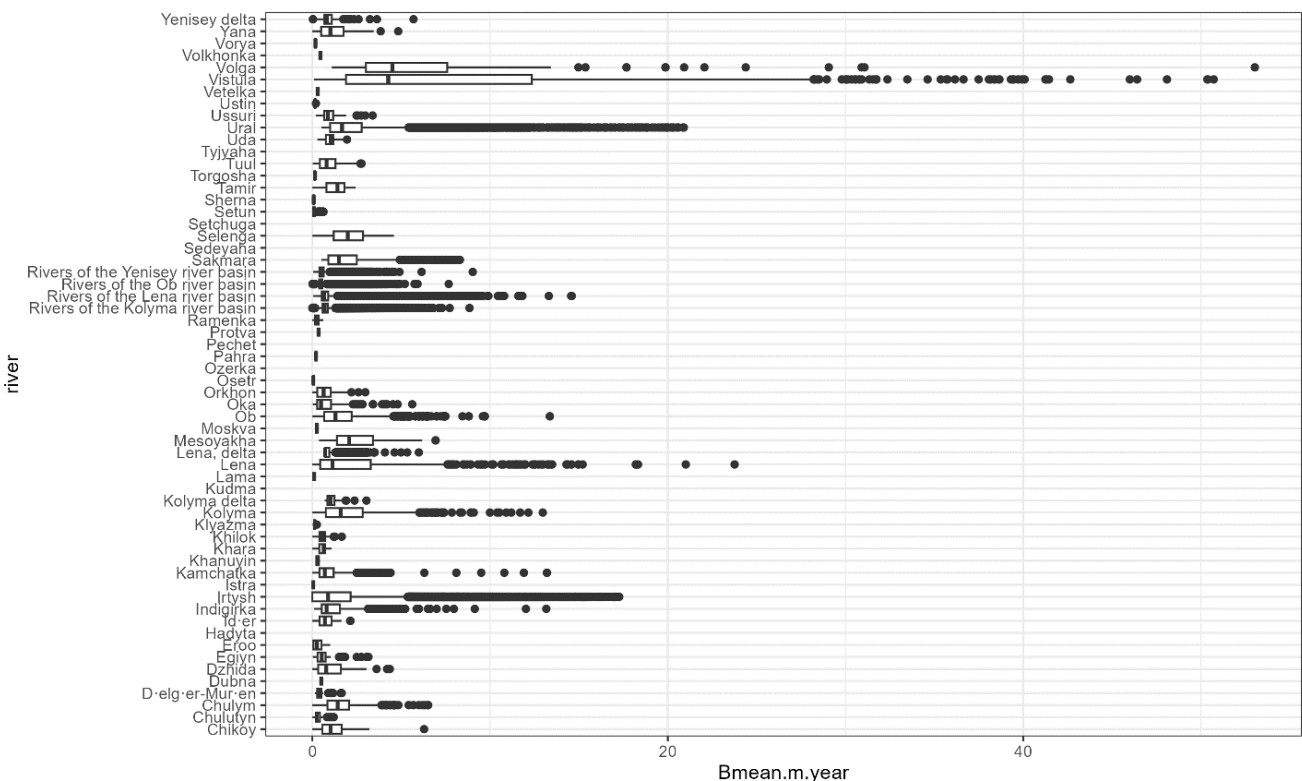

**Fig. 10. Distribution of mean annual bank retreat values by rivers**

The largest values of maximum bank retreat are for the Ob, Kamchatka and Indigirka Rivers. Values of eroded areas due to bank retreat (Amean) for whole rivers banks are in the ratio between 0.01 and 226,000 m$^2$/year. This parameter was estimated for 15 rivers that were processed by methods II, III and V. The mean value of this maximum bank retreat is 913 m$^2$/year; the median values is 46 m$^2$/year. The quartile of 75% has a value of 130 m$^2$/year, and its 25% is 28 m$^2$/year.

Values of mean annual bank erosion volumes (Vmean) for the banks of whole rivers are in the range between 0.01 and $11 \cdot 10^6$ ton/year. This parameter was esteemed only for eight rivers that were processed by method III. The mean value of Vmean is 17,000 ton/year; the median value is 889 t/year. The quartile of 75% has a value of 2,500 ton/year, and its 25% is 407 ton/year. The largest values of maximum bank retreat are for the Lena and Kolyma Rivers.

Significant total length of the analyzed sections allows retreat rates to be assessed in a statistical way by estimating the recurrence of different scour rates along the length of the eroded sections. Their total length for the studied rivers ranges from 24 to 49% of the total length of the banks, reaching maximum values at sections with completely unrestricted meandering. Resulting percentile curves shows the comparative rarity of extreme bank retreat rates (Chalov and Shkolnyi, 2018), which, at the same time, determine the main portion of eroded areas and sediment source in the considered rivers (figure 11).

The combined length of the sections studied allows for the assessment of retreat rates through statistical analysis, which entails determining the frequencies of different retreat rates along the eroded sections. Their total length for the studied rivers




comprises between 24 and 49% of their total length along their banks, with the highest percentages being found in sections with completely unrestricted meandering. The resulting percentile curves demonstrate the comparative scarcity of extreme bank retreat rates (Chalov and Shkolnyi, 2018), which, concurrently, account for the majority of eroded areas and sediment sources in the studied rivers (figure 11).

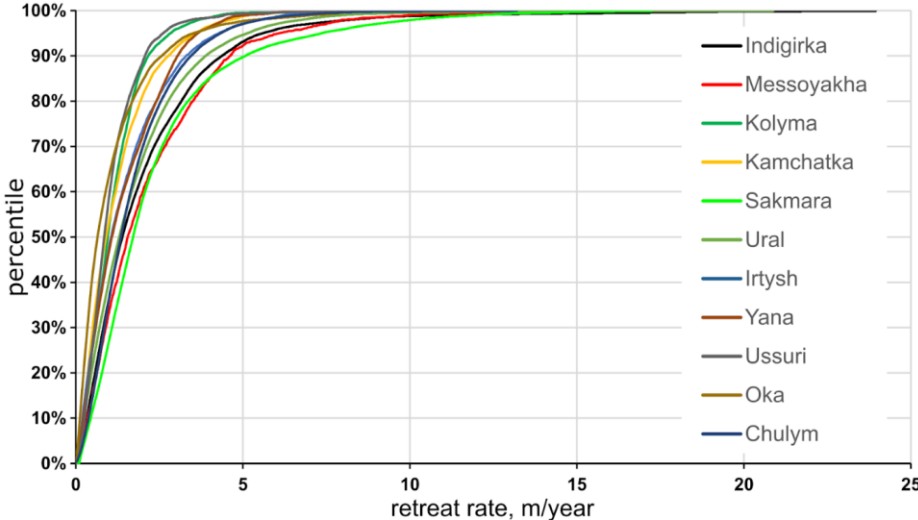


**Figure 11. Percentile curves of the retreat rates for the observed rivers' eroding sections**

## 4. Discussion

### 4.1. Drivers of riverbank migration across Northern Eurasia

The presented dataset is different from previous products in that it contains information on small, medium and large rivers
alike. By attributing mean bank retreat rates to annual discharges (Fig. 12), our results confirms that size is the first-order control on riverbank erosion at large scales following previous estimates of large rivers (Langhorst and Pavelsky, 2023). The general relationship of *Bmean* and annual discharge *Q* is explained by quadratic law: $Bmean=f(Q^2)$. At the time influence of river size control is rather as far as riverbank erosion is complex, with many different processes and mechanisms depending on multiple forcing parameters working in tandem and varying both spatially and temporally. Among them, channel patterns
and stream geometry, bank composition, water temperature and soil moisture, which all impact both separately and jointly to bank erosion rates, and related to also climatic, geological and other drivers.





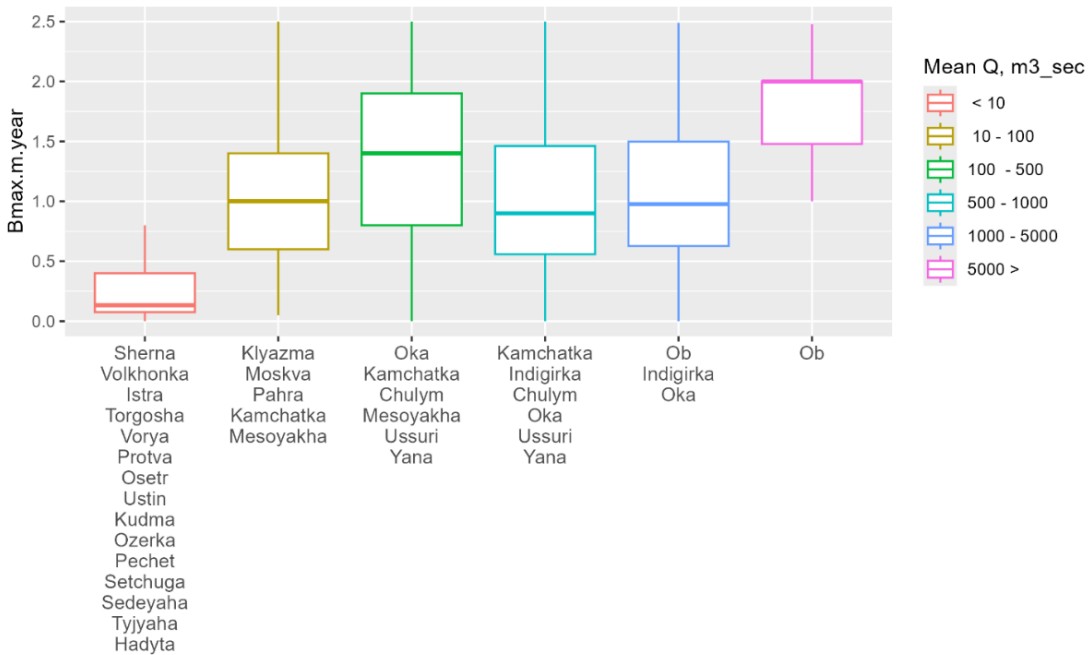

**Fig. 12. Boxplot of reach-averaged bank retreat rates with respect to river size (annual discharge $Q$ [m³/s])**

The first order capturing these drivers' influence is specific to each catchment or river. The NERM dataset provides sufficient

information to assess the impact of catchment and longitudinal changes on bank-erosion mechanisms. Such an example is

provided by the bank migration of the Ob River, which is the longest river in Northern Eurasia, with an average *Bmean* of 2

m/year⁻¹ (Fig. 13a). The mean annual erosion rate of banks in the upstream Ob is 2.4 m year⁻¹, while the maximum annual

erosion rate is 26.3 m year⁻¹. In the sections of the Ob River located downstream from the Novosibirsk reservoir, erosion rates

decrease (mean annual erosion rate is 1 m year⁻¹, maximum annual erosion rate is 6.6 m. year⁻¹). The latitudinal section of the

Ob River (from the confluence of the Ob River and the Vakh River to the confluence of the Ob River and the Irtysh River)

shows an increase in average rates of bank migration to 2.5 m year⁻¹, and a maximum rate of erosion of 16.4 m year⁻¹.

Downstream Ob shows similar riverbank migration rates as at the latitudinal section of the Ob River (mean annual erosion rate

is 2 m year⁻¹, maximum annual erosion rate is 17 m year⁻¹). The area of erosion on the Ob River has increased from almost 4

million square meters in the sections located downstream from the Novosibirsk reservoir to over 120 million

square meters on the downstream Ob (Fig. 13b).





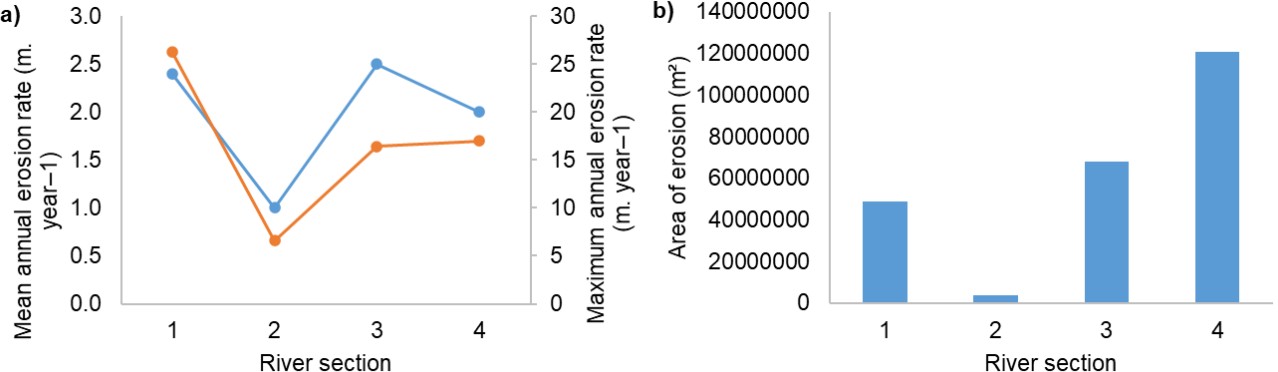

**Fig. 13. Riverbank erosion of the Ob River: a – erosion rate, b – area of erosion. 1 – Upstream Ob River, 2 – Middle stream of Ob River (from the Novosibirsk reservoir to the confluence of the Ob River and the Tom River), 3 – Middle stream Ob River (from the confluence of the Ob River and the Vakh River to the confluence of the Ob River and the Irtysh River), 4 – Downstream Ob River (from the confluence of the Ob River and the Irtysh River to the city of Salekhard)**

The mean rates of bank retreat are modeled using a gamma distribution due to the presence of extreme rates of riverbank migration.Extreme bank erosion on large Northern Eurasia rivers is frequently more rapid than average rates due to certain factors, such as low-density material like pyroclastic sand found on the Kamchatka River (Chalov et al., 2021) or permafrost layers as reported by (Gautier et al., 2021). The greatest rates of platform change are typically found on large rivers featuring complex braided channels. The spatial analysis of erosion rates along rivers reveals extreme bank retreat rates that significantly exceed the 95[th] percentile. On average, the river's retreat is typically around 2% of its width annually, with a yearly rate of between 2 and 15 meters for the larger rivers in the study. In the area surrounding the Partizan settlement, near the Lena River, extreme values of long-term measurements reach 35 meters per year. This is particularly evident where discharge from various branches converges into a single channel, causing a significant bend with a curvature ratio of approximately 2. The Indigirka River at Sypnoy Yar exhibits annual retreat rates of up to 24 meters; in this location, the receding right bank of the river is a 30-meter high plateau ledge formed from frozen sands. In the two described regions, erosion results in the delivery of several million tons of sediment annually, which in turn leads to the formation of riffles downstream and poses challenges for navigation. The rate of bank erosion on the Kamchatka River is not as extreme as some others, with a maximum annual retreat of up to 7 meters, but it can still cause the river width to decrease by 10% or more each year, resulting in the quick movement and cyclical meanders cutoff.

The NERM dataset provides information on other drivers at the scale of Northern Eurasia – e.g., it also shows the importance of climatic impacts which can be seen at graphs dividing retreat rates on climate zones by Koppen classification and latitudes (Fig. 14). Mean annual bank retreat rates are decreasing from south to north (from 1.9±0.8 m/year within the 40–50° zone to 0.5±0.2 m/year within the 70–75° zone).




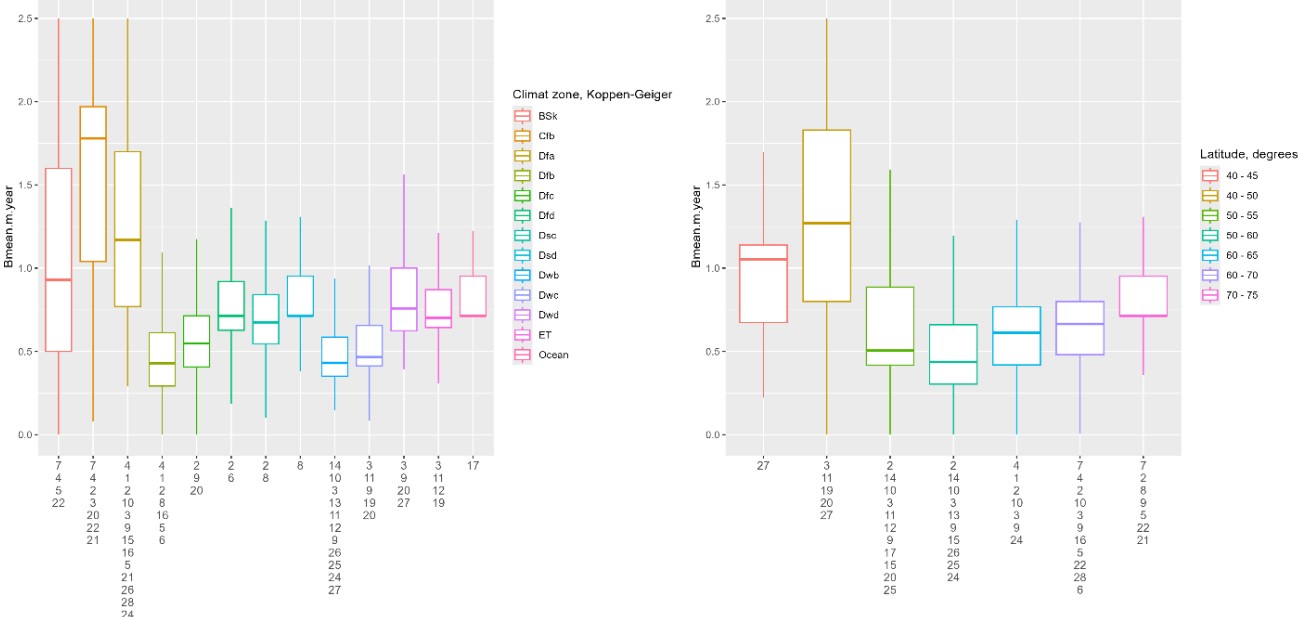

**Fig. 14. Boxplot of reach-averaged bank retreat rates with respect to climate zones by Koppen classification (see explanations in the text) and latitude**

555  Another natural driver that plays an important role in the channel evolution of Northern Eurasia rivers is permafrost (Rowland et al., 2010). The distribution of bank erosion rates in different permafrost zones reflects contrasting impacts of permafrost on riverbank migration. The average rate of bank retreat (*Bmean* [m/year]) varies in NERM from 1.3±0.8 m/year in the permafrost-free zone to nearly 1 m/year where permafrost exists. By contrast, the areas of riverbank retreat (A [m$^2$/year]) increase with the increase in permafrost distribution (Fig. 15). This enables to conclude that thermal erosion in combination

560  with mechanical erosion determines the greater susceptibility of riverbanks to destruction in the permafrost distribution zone, but the erosion rates of banks composed of permafrost soils are lower due to the soil adhesion mechanism. This thesis is generally in line with Rowland et al. (2023), who found for the past that erosion rates in permafrost-affected rivers were on average nine times lower than in non-permafrost-affected systems.





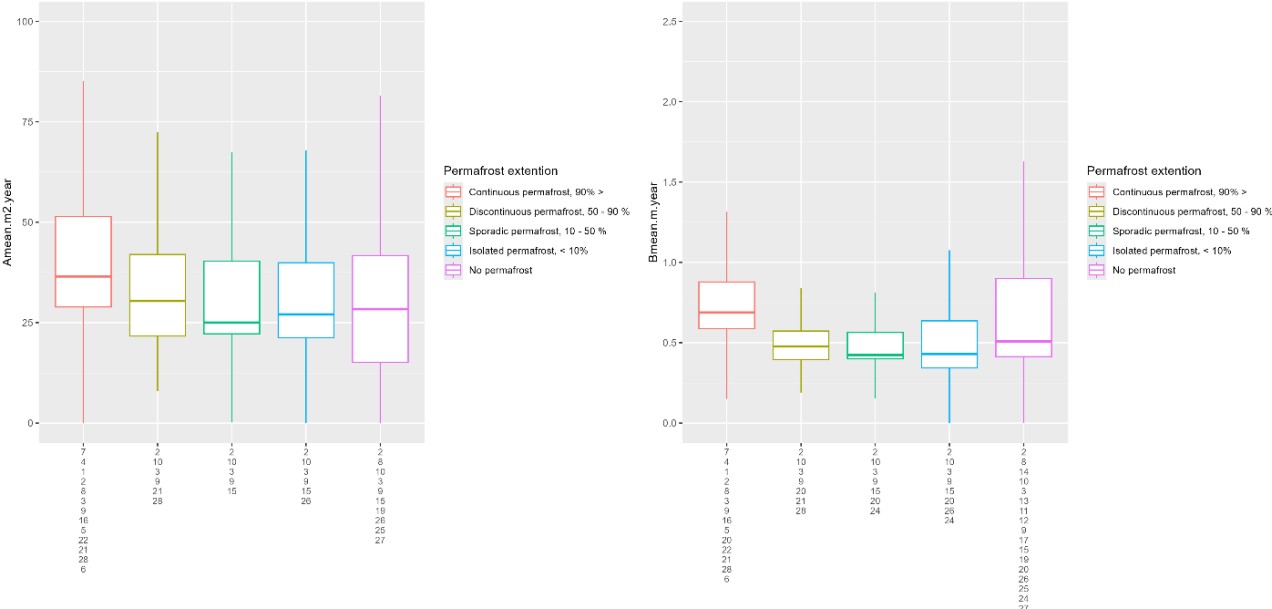

**Fig. 15. Boxplot of reach-averaged bank retreat rates with respect to permafrost (left – bank areas destroyed by bank erosion; right – average rates of bank erosion)**

These finding related to permafrost impact are confirmed at permafrost-affected rivers where data of channel changes was extended from 1950 to 1970 with the use of manual digitizing by Keyhole images. In the Lena River delta (#8), the highest bank migration rates are related to the ice complex (or Yedoma) areas. Specifically, up to 15% of the bankline may be eroded in the downstream of the Trofimovskaya and Olenekskaya branches (Fig. 16). The highest mean annual erosion rates occurred in yedoma on the Sobo-Sise Island with erosion area amounted to 0.58 km$^2$ over 21 years. The mean annual erosion rate is 4.74 m year$^{-1}$, and the maximum rate increases to 15 m year$^{-1}$. Due to a rise in air temperature rise from 0.86 °C per decade from 1979 to 2021 to 1.61 °C per decade during 2000–21 (Chalov et al., 2023a; Gelfan et al., 2017), there was an increase of 1.95 m year$^{-1}$ (or 95%) in 2000–21 compared to 1964–2000 (Fig. 16). The most significant increase in erosion rates (3 times between periods) is observed in the Olenekskaya branch and is also related with the ice complex on Kurungnakh Island. It is important to note that along ice-wedged complexes, the high rates of bank retreat remain stable (Sobo-Sose Island). Previous studies on channel migration for the Lena River Delta concentrate on the yedoma permafrost cliff on Sobo-Sise Island at Sardakhskaya branch (Fuchs et al., 2020). The cliff length is 1,660 m and the vertical heights are up to 30 m above mean river water level. The authors manually digitized the upper cliff line on the images from 1965 to 2018. Erosion rates vary from 4.8 to 15.7 m year$^{-1}$ in different parts of cliff, and the mean annual erosion rate is 6.1 m year$^{-1}$.



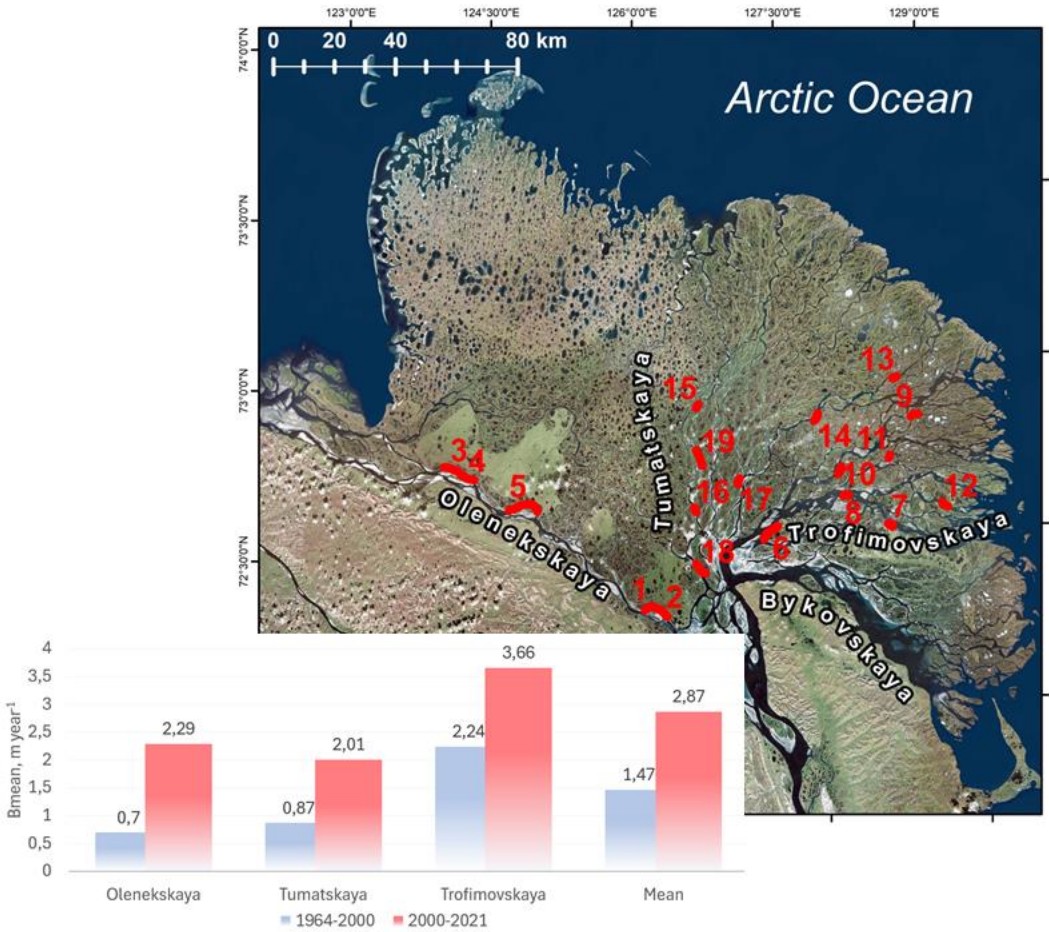

**Fig. 16. Bank migration rates over Lena delta: location of riverine reaches with estimates (on the top, reaches are indicated by numbers) and values for different time spans (on the bottom) (© Esri Imagery)**

Similar patterns are observed to those of the downstream Kolyma River, where a significant increase in riverbank migration rates was observed both upstream from the delta and within the delta. Similar climatic drivers here influence both the stability of ice-wedge complexes widely distributed along channel banks (Murton et al., 2015; Szumińska et al., 2023) and an increase in riverbank migration rates. It is interesting to note that the rates of channel migration were significantly higher within the delta during all considered periods (Fig. 17).




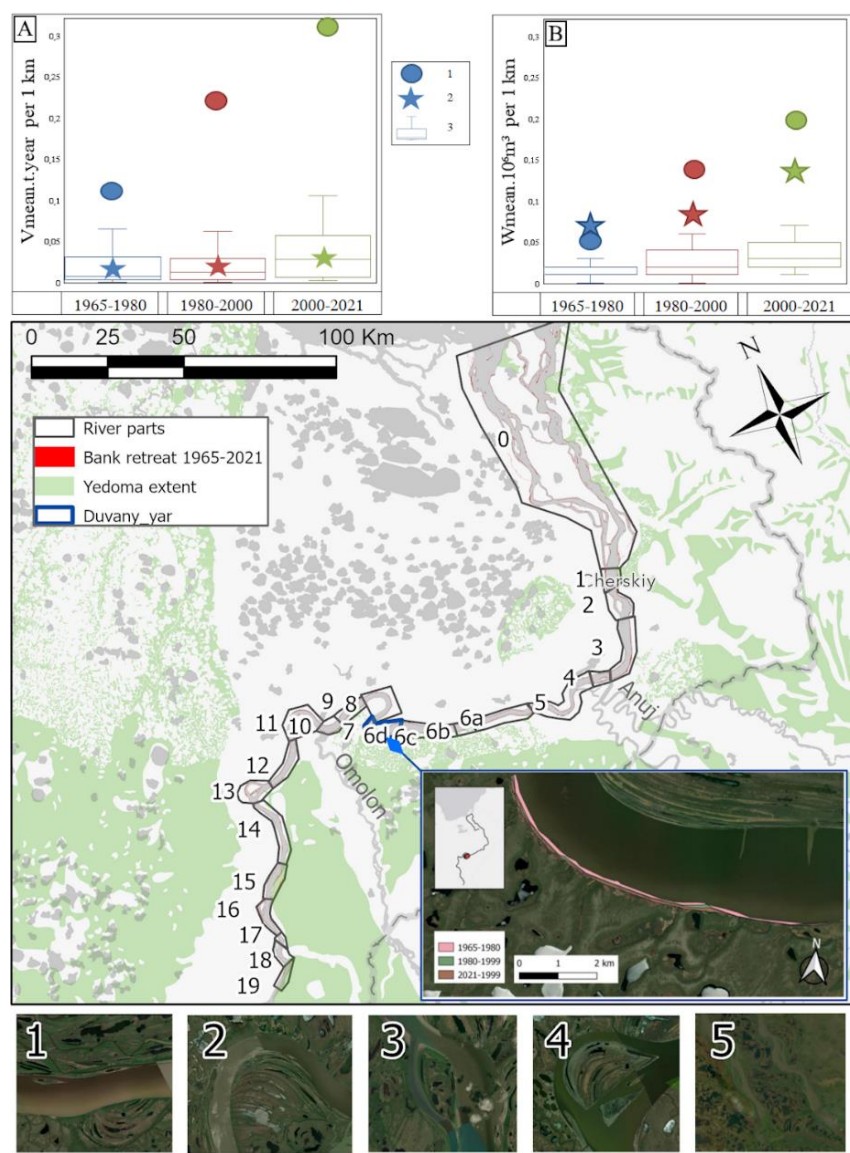

**Fig. 17. Boxplot comparing the distribution of volume of bank retreat (left) and sediment mobilization due to bank retreat t/year (right) in 1965–80, 1981–2000 and 2000–21 in Kolyma upstream from delta (boxplots) and within delta (dots, reach 0 on the map). Stars indicate reach 6 on the map (Yedoma Duvanny Yar complex) (© Esri Imagery)**

### 4.2. In-channel erosion contribution to sediment yield

The total volume of channel erosion is estimated at 151 million tons per year in the 1,680-km section of the lower Ob, 15.9 million tons per year in the 1,500-km section of the Yenisei, and 338 million tons per year in the 1,800-km section of the Lena. In the Selenga and the Kamchatka, the volume of channel deformations is comparable to the values typical for the lower course of the Yenisey River. On the Kamchatka River, the total eroded area during the comparison period (from 23 to 50 years, depending on the availability of satellite images for river sections) was 23.5 km$^2$, which corresponds to an average volume of





channel deformations of 670,000 m²/year. Thus for the Kamchatka, ~4.6 million tons of sediments per year enter the channel due to bank erosion. In some years, this value may increase because of the bends cutoff.

Sections of a braided channel exert the greatest influence on sediment runoff. In the Selenga basin, the maximum values of bank retreat (on average at the tops of bends from 9 to 16 m/year) are mainly situated in the section of the braided channel. At the same time, in the sections of the channel with wide floodplains, more than half of the entire area of bank erosion is related

to the erosion of river islands. On the Lena River section from Pokrovsk to Zhigansk cities, 60% of the total volume of bank erosion is related to island erosion (11 thousand m2 year-1 km-1), and it is 40% to erosion of the bedrock. The left and right banks have an equal ratio of 20% to each other (3.7-thousand m2 year-1 km-1). The maximum intensity of bank erosion is associated with the incision of river valleys and sand bars. On all rivers, the ratio of the intensity of bank erosion of the right and left banks is approximately the same. For the left bank of the Kamchatka River, the length of the eroded banks is 37%,

and for the right bank, it is 44%.

These combined results offer significant insights into the sediment budget of large river basins. In the lower reaches of the world's biggest rivers, the quantity of channel erosion debris that enters is roughly the same as the amount of sediment being washed away, and materials eroded from riverbanks are highly connected to the channel. Based on this, it can be inferred that the channel component of sediment runoff exhibits latitudinal zonality and is influenced by large-scale factors, which in turn

are affected by river size. In the Ob River's middle and downstream sections (below where it meets the Vakh River), approximately 33.5 million tons of sediment enter the channel yearly due to channel deformations, which is lower than the sediment load at the river's mouth. The sediment influx into the channel due to bank erosion within the river basins of southern rivers, where unstable channels are common, often far surpasses the sediment discharge of rivers, as seen in the Selenga basin where it is 11-fold (Table 5). In the foothills, sediment runoff originating from rivers achieves high levels, with significant

disparities in the extreme mitigation measures taken for channel control. In the Kamchatka River basin, the river itself carries two times more material eroded from its banks than is transported downstream in the sediment runoff, with at least half of this material accumulating in the riverbeds.

**Table 5. Calculations of volume of material entering channels of the Selenga River and its tributaries as a result of bank erosion (within the main channel or main branch)**

| River | Total volume, m³/year | Total volume Ton/year* |
|---|---|---|
| Dzhida | 847,198 | 1,440,200 |
| Delger mörön | 233,282 | 396,600 |
| Ider | 412,960 | 702,000 |
| Orkhon | 1,143,891 | 1,944,600 |
| Selenga | 7,304,712 | 12,418,000 |
| Tamir | 448,632 | 762,700 |





| | | |
|---|---|---|
| Tuul | 684,115 | 1,163,000 |
| Uda | 454,782 | 773,100 |
| Hanuyin | 4,371 | 7,400 |
| Hara | 142,079 | 241,500 |
| Khilok | 509,027 | 865,300 |
| Chikoy | 1,577,407 | 2,681,600 |
| Chuluut | 67,734 | 115,100 |
| Egiin Gol | 396,272 | 673,700 |
| Eroo | 25,834 | 43,900 |
| Totally from the catchment | 14,252,297 m$^3$/year | 24,230,000 t/year |
| Accounting erosion of islands | | 26,700,000 t/year |

## 5 Data availability

The presented datasets are available open access via the ZENODO repository (https://doi.org/10.5281/zenodo.11072919) (Chalov et al., 2025).

## 6 Conclusions

The NERM dataset offers the comprehensive riverbank migration assessment for particular areas. We utilised a unified dataset to examine statistical data on bank erosion, river discharge, and catchment factors, including permafrost extent and natural zone, using multiple analytical tools. River size was discovered to be a key factor in riverbank erosion. Confirmation of the role of secondary controls in Northern Eurasia has been established, encompassing permafrost distribution and diverse climatic/natural zones. NERM serves as a case study to refine and verify theoretical models, offering insights into sediment origins in river systems by integrating riverbank erosion rates with sediment mobilization from eroded riverbanks. The comprehensive dataset presented offers a potential insight that further research employing multi-statistical methods could reveal quantitative laws governing riverbank migration at regional or catchment scales, influenced by geological, hydrological, and climatic factors.

## Author contributions.

SRC: Conceptualization, Investigation, Methodology, Supervision, Writing – original draft preparation, Writing – review & editing; VI: Conceptualization, Formal analysis, Methodology, Project administration, Investigation, Visualization, Writing –



original draft preparation, Writing – review & editing; DS: Investigation, Methodology, Writing – original draft preparation, Writing – review & editing; EP: Project administration, Writing – review & editing, Investigation; MH: Investigation, Writing – original draft preparation; AK: Investigation, Writing – original draft preparation; RSC: Investigation, Supervision; DB, AC, PG, AK, RK, UK, NM, ET, KP, AZ, RA, AV, LT, AT and DF: Investigation. All authors reviewed the manuscript.

**Competing interests.**

The authors declare that they have no conflict of interest.

**Disclaimer.**

The data are provided with no warranty.

**Acknowledgements.**

This work was supported by the grant of the The Government of the Russian Federation (Agreement №075-15-2025-008 date 27.02 2025) and the state assignment of the Makkaveev laboratory of Soil Erosion and Fluvial Processes, Faculty of Geography, Lomonosov Moscow State University (CITIS 121051100166-4).

The authors used Paperpal AI Microsoft Word plugin and Trinka AI Writing and Grammar Checker Tool for English grammar improvement.

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
