# Peer review of "Multi-tool dataset on Northern Eurasian Riverbank Migration (NERM)"

_Earth System Science Data, 2025_

## Author Response (AR1)

Dear Editor,

We are very grateful for the overall positive comments, confirming that our work is of interest, which was a great motivation for us to improve the paper further. We have agreed with most of the Reviewers' comments. Following their recommendations and constructive feedback, we carefully revised the manuscript and some of the figures and tables.

We believe that our efforts improved the new version and we hope that it could now be considered for publication in the ESSD journal. All main changes made are additionally highlighted in yellow in the revised version so as to facilitate the revision process.

We look forward to hearing from you soon,

The co-authors

**Reviewer #1:**

This is a significant research work presenting a multi-tool dataset of channel or river-bank erosion rates (NERM) acquired from 12 major river basins covering 250 rivers across Northern Eurasia, covering around 140,000 km2 over a 70-year span. The NERM dataset is based on integrating in-situ assessments, remote sensing data, and geospatial applications, through which the dynamics of river-bank erosion under different hydro-climatic and geomorphic conditions are modelled accurately. This dataset has underlined the research gap by emphasizing the data scarcity regarding riverbank erosion of North Eurasian rivers. Hence, the issue of data scarcity has been addressed in this work by combining the hydro-climatic and geomorphological parameters such as water discharge, permafrost zones, channel patterns, and sediment estimates. The NERM dataset efficiently addresses the research gap by providing riverbank erosion estimates using multiple techniques. It is important to erosion monitoring in long-term and large-scale comparative analyses. The inclusion of large river-deltaic systems (Kolyma, Ob, Lena, and Yensei), medium, and minor riverine systems like Sedayakha, Tyjakha, and Khaduta, as well as many other rivers across North Eurasia, within the dataset enhances the spatial heterogeneity and comparisons across climatological and geological settings.

The methodology adopted in this work for riverbank assessment is recommendable because it efficiently combines techniques, such as in-situ assessments, UAV monitoring, area-based approaches, satellite imagery analysis, GIS digitizing algorithms, and automated statistical methods. These techniques have enriched the NERM dataset and provided important insights regarding river-bank erosion dynamics, which establishes a standard for erosion-based assessments. By generating river-bank erosion estimates using different methods, this study highlights the insights and importance of site-specific deployment of methods for future erosion assessments. In addition, the systematic validation of area-based and erosion-based methods of riverbank migration estimates, within considerable error margins, highlights the credibility of the NERM dataset. Although the methods, particularly remote sensing and geospatial approaches, frequently come across several issues in delineating river bank erosion, due to variables such as forest cover, vegetation patches, shadows, and complex relief features, etc.

The techniques used for generating results are further enhanced by using the spatial-temporal comparisons and gamma distributions for the large river-deltaic systems like Lena Ob and Lena. Additionally, it quantifies thermal erosion from permafrost and climate impacts, providing insights into future projections of riverine dynamics. Furthermore, the discussion part involves the analysis of several factors inducing riverbank migration across the river basin of Northern Eurasia. Then this work becomes immensely significant due to the development of interactive web platform GISCARTA developed to provision access to the NERM database, which is enabling its visualization and acquisition of key outputs of this study such as bank erosion/ retreat rates, erosion area, sediment yield etc in accessible file formats. The user-friendly interface of the GISCARTA platform' with regular updates is essential for researchers to monitor and analyse the fluvial processes across the riverine systems of different spatio-temporal scale.

It can be concluded by stating that this research work is highly recommendable due to the incorporation of the river basins of different spatiotemporal scales and assessing the scenario of riverbank erosion via suitable methods, which will give ideas to other researchers to apply site-specific methods for assessing riverbank erosion. It is noteworthy that different estimates have been completed to form a database or dataset of NERM, which is promoting advances in the domain of fluvial geomorphology by being an interactive source in the form of GISCARTA platforms, providing access to the estimates generated through detailed, multi-scale assessment of riverbank erosion processes across Northern Eurasia.

**Response:** We sincerely thank the reviewer for their thorough and insightful evaluation of our manuscript. We greatly appreciate the recognition of the significance of the NERM dataset, the integrative methodological framework, and the potential contribution of our work to addressing data scarcity in riverbank erosion studies across Northern Eurasia.

We fully agree with the reviewer that the challenges associated with riverbank delineation—such as vegetation, shadows, and terrain complexity—require methodological flexibility. Therefore, one of the core aims of our work has been to highlight the value of using a multi-method approach adapted to specific site conditions, which, as the reviewer notes, may serve as a guide for future studies.

Once again, we appreciate the reviewer's positive and detailed feedback. It confirms the relevance and potential of our dataset and web platform in advancing large-scale, long-term erosion research and fostering future applications in fluvial geomorphology.

**Reviewer #2:**

I enjoyed reviewing this relevant article that presents a new dataset of riverbank migration rates for Eurasian rivers at a multi-scale, and that applies and compares multi-methodological approaches, such as remote sensing techniques, and some areas with field control. The combination of migration rates and a geographic-environmental perspective is the main component for river classification. I have some observations and general comments on shape and content that are intended to improve the message of the manuscript and its clarity to the reader.

**Response:** We thank the reviewer for the positive feedback and for the careful attention to our study. We are pleased that you found the dataset on riverbank migration across Eurasia relevant, and that you recognized the value of our multi-scale, multi-methodological approach, which integrates remote sensing techniques with field validation.

We fully agree that combining migration rate estimates with a geographic and environmental perspective is essential for meaningful river classification. This principle guided both the structure of our dataset and the analytical framework applied in the study.

We also greatly appreciate your constructive observations and general comments regarding the shape and content of the manuscript. We have carefully considered each of them and made the necessary revisions to improve the clarity, coherence, and readability of the text.

Thank you again for your thoughtful review and for contributing to the improvement of our work.

Style: It isn't very clear for the reader to understand the analysed sections of the rivers. The authors assessed 140,000km of rivers, but why appear only a few lines and areas indicated with squares and rectangles in the map, which seems to be smaller than the 140,000km in length? Figure 1 deserves a better/expanded caption. What do the squares, lines, etc., mean?

**Response:** Thank you for pointing this out. We have revised Figure 1 to improve its clarity and visual representation of the analysed sections. The updated version includes a clearer indication of the studied river segments and the logic behind the selected areas. We have also expanded the figure caption to explain the meaning of the lines, squares, and rectangles, making it easier for the reader to understand the spatial distribution and scale of the data analyzed.

For Table 1 also explain better. For example:

- Yakutsk, length 1750km. What is this length? From upstream to downstream until Yakutsk? From Yakutsk downstream? It is recurrently unclear what the specific studied areas, reaches, and the averaged numbers discussed in the text etc.

- Rivers of the Lena basins, etc. What rivers are those? Why are those rivers not plotted in a map 1 with some differential colour?

- The Ob, for example, is also confusing. We have three areas/lines in the Obi basin, but only one text in Table 1.

Organize table 1 concentrating on the continuity of the information per basins (ex: Ob, Rivers of the Ob basin, Irtysh, Chulym), and not scattering information the same river basin in different sections of the table. Please use the same organization in Figure 10 and expand the caption of Figure 10.

**Response:** Thank you for this valuable suggestion. It is important to note, that partly detailed description of the river network was presented in the text, however both figure 1 and table 1 were adjusted to add whole necessary information regarding rivers included into database. We have reorganized Table 1 to improve the continuity of information by grouping rivers according to their respective basins. To further enhance navigation and clarity, we added a new "Region" column and adjusted the order of rivers from west to east. We also expanded the "Rivers" column to include more precise descriptions of the analyzed sections. Similarly, we updated Figure 10 to follow the same basin-based structure.

In Figure 5 a methodological example is presented using a reach of the Ob River. It is an anabranching reach with changes on the lateral banks and on the banks of islands. However, the text mentions that only the lateral banks are computed…"for each reach expressed, the right- and left-bank migration polygons were divided by the total surface (what are is that? it is water surface area?? Eroded area on the lateral banks or all the eroded polygons? Explain it better,… And…"then by the length of the oldest of the two channel banks and then by the number of years of the analyzed time interval". It is okay for me, but because the figure focuses on the changes by erosion, it is important to improve the explanation of the calculation and the figure itself. Indicate the polygons used in the figure for the computation, and indicate which polygons are not included in the calculation. Add a mathematical expression (formula) to the written explanation of the computation. Also include a good legend or write in the caption what green, red, etc, means on the left figure and what yellows, greens, etc, mean in the right figure. Expand the caption of the figure.

**Response:** Thank you for your detailed and constructive comment. We have carefully revised both the text and Figure 5 to address your concerns. Text was corrected - we analyzed also islands banks, not only left and right. Migration rate calculation were checked. Figure was updated with legend.

Figure 6- Include coordinates in the figure and North.

**Response:** North arrow was included; caption was updated.

It is mentioned in the test that Table 3 is related to calculations in the Irtysh River. Include that information and expand the caption of Table 3.

**Response:** Text and caption were updated.

Sediment Yields: I assume this is a question of terminology/meaning used in different countries that can be confusing for readers. For the majority of geomorphologists, sediment yield is the amount (weight or volume) of sediment reaching or passing a point of interest in a given period of time, normalized by unit of drainage area. Thus, it is normalized in function of the drainage area that sources the sediments (X unit of weight/unit time/unit of area. However, in this study, the focus is on the source of sediments by lateral erosion per unit of channel length per unit of time. Thus, I suggest changing the term " Sediment yield" by using the terminology " gross bank-erosion sediment yield". The incorporation of the term "gross" is important because the authors are not presenting the net bank erosion sediment yield, but, apparently, just the calculation of erosion. The net sediment yield is the difference between erosion and sedimentation per unit of length, and in many rivers and fluvial reaches of rivers, the balance can be positive (+) or, in others, the bank sediment yield negative (-).

I consider it important for the authors to redefine the definition of sediment yield and make clearer these conceptual issues in the methods and throughout the paper.

**Response:** The terminology related to sediment yield was also revised throughout the text. Definitely, we did not analyse net erosion, and we used reviewer suggestion with the proper terminology as "gross bank-erosion sediment yield" as a very reasonable. We checked throughout the text definition and terminology of sediment yield.

River Classification: the main factors here are geographic-climatic (presence of permafrost and bioclimatic zones (rainforest, Arid desert, steppe, savanna etc). I understand that it is difficult to synthesize such an amount of information in a single paper and that the authors chose those variables for the classification, but I consider that it would be important to clarify to the readers, in a short sentence, the limitations of that approach and why they chose that option. Thus, I consider it is relevant to include a short sentence in the text explaining that beyond the main factors they considered, such as permafrost and conditioning factors of the migration rates in a short scale (because just a few decades can be assessed due to the limitation of satellite products), channel patterns and the geotectonic context of the basins

would be a desirable factor also to be considered. My point here is that it is desirable to highlight to the readers in a short sentence that although they chose an specific approach/way to classify the rivers, the results are indicative of average rivers´ behavior and that the diversity of migration rates along the rivers can be variable and, ideally, furtherly assessed for geomorphological reaches and channel patterns, and not just averaged over long distances. It is important because significant errors are introduced when normalizing budgets per unit of length using only lateral bank records over long distances.

**Response:** We definitely agree with the point which states that proposed classification of channel drivers is not full. We believe that presented database is a part of broader process which will involve channel pattern classification and its factors in larger extent. While compiling this database, we were technically limited to those drivers which are available to analyse, while channel patterns delineation is currently prepared along different regions of Northern Eurasia and this is the process which is under preparation. As an example, you can see some papers on the topic which are currently published on the particular regions (https://doi.org/10.35634/2412-9518-2024-34-3-308-314 ; https://doi.org/10.17072/2079-7877-2023-1-100-115 ). Here the geotectonic context is a part of channel pattern classification, mostly in relation to floodplain width, incised/free conditions of channel development. In the current database it is not possible technically to add detailed channel pattern classification, while we used the map: "Channel morphology regime of rivers of USSR" (Chalov et al., 2018) to attribute reaches to some of the channel characteristics (see part 2.5) . We add these considerations in the text as well.

General observation on figures: The captions of the figures need to be improved. Provide more details on the figure content, variables etc. Usually, the captions do not provide enough information to the reader.

**Response:** We have revised and expanded the figure captions throughout the manuscript to provide more detailed descriptions of the figure content, variables used, and their relevance to the analysis.

Conclusion: This is a valuable article with new results and a relevant database. I am confident the authors can consider my comments and suggestions.

**Response:** We sincerely thank the reviewer for the positive assessment of our work and for recognizing the value of the results and the NERM database. We greatly appreciate your thoughtful comments and suggestions, all of which have been carefully considered and incorporated to improve the manuscript.

---

## Author Response (AR2)

Dear Editor,

Thank you for your positive evaluation of our manuscript and for the constructive feedback. We are pleased to hear that our work has been accepted for publication pending minor revisions. Below, we outline the steps we have taken to address your comments.

**Comment:** Please update the metadata of your data publication by including all ORCIDs of the authors.

**Response:** We have updated the metadata of the Zenodo data publication to include all ORCIDs of the authors.

**Comment:** Include in your zenodo data publication a technical readme, a technical description of the dataset, e.g. the content of the technical fields and the abbreviations.

**Response:** A detailed technical README file has been added to the Zenodo dataset, clarifying the structure and content of the dataset fields and definitions of all technical terms and abbreviations used.

**Comment:** Check for abbreviations in your main manuscript, the majority of abbreviations is well introduced. In the introduction, please consider to explain GLAD.

**Response:** We have added an explanation of GLAD (Global Land Analysis and Discovery) in the Introduction to improve clarity.

Comment: Please ensure that the color schemes used in your maps and charts allow readers with colour vision deficiencies to correctly interpret your findings. Please check your figures using the Coblis – Color Blindness Simulator (https://www.color-blindness.com/coblis-color-blindness-simulator/) and revise the colour schemes accordingly with the next file upload request. -> Fig. 5, 11

**Response:** We have checked Figures 5 and 11 using the Coblis Color Blindness Simulator to ensure accessibility for readers with color vision deficiencies.

We confirm that all requested changes have been implemented, and the revised files (manuscript and dataset) are now fully compliant with your recommendations. Please let us know if any further adjustments are needed.

Once again, we sincerely appreciate the time and effort invested by you and the reviewers in evaluating our work.

We look forward to hearing from you soon,

The co-authors